# Speculative Jacobi-Denoising Decoding for Accelerating Autoregressive Text-to-image Generation

**Yao Teng**[1]   **Fuyun Wang**[2]   **Xian Liu**[2]   **Zhekai Chen**[1]   **Han Shi**[3]
**Yu Wang**[4]   **Zhenguo Li**[3]   **Weiyang Liu**[2]   **Difan Zou**[1]   **Xihui Liu**[1][*]
[1]The University of Hong Kong   [2]CUHK   [3]Huawei Noah's Ark Lab   [4]Tsinghua University

## Abstract

As a new paradigm of visual content generation, autoregressive text-to-image models suffer from slow inference due to their sequential token-by-token decoding process, often requiring thousands of model forward passes to generate a single image. To address this inefficiency, we propose Speculative Jacobi-Denoising Decoding (SJD2), a framework that incorporates the denoising process into Jacobi iterations to enable parallel token generation in autoregressive models. Our method introduces a next-clean-token prediction paradigm that enables the pre-trained autoregressive models to accept noise-perturbed token embeddings and predict the next clean tokens through low-cost fine-tuning. This denoising paradigm guides the model towards more stable Jacobi trajectories. During inference, our method initializes token sequences with Gaussian noise and performs iterative next-clean-token-prediction in the embedding space. We employ a probabilistic criterion to verify and accept multiple tokens in parallel, and refine the unaccepted tokens for the next iteration with the denoising trajectory. Experiments show that our method can accelerate generation by reducing model forward passes while maintaining the visual quality of generated images.

## 1   Introduction

Autoregressive models have emerged as a cornerstone of visual generative tasks through next-token prediction [1–4]. However, the autoregressive paradigm suffers from significant inference latency due to its sequential, token-by-token decoding process. For instance, generating a single high-resolution image often requires thousands of sequential forward passes. To address this challenge, we focus on accelerating autoregressive text-to-image generation models via parallel token decoding.

Jacobi decoding [5] is an iterative method that accelerates the inference of autoregressive models through parallel token decoding without any training. This method operates on a sequence of randomly initialized tokens and iteratively calls the neural network to refine the tokens until the *convergence* (*i.e.*, tokens are correctly decoded). Its variant, Speculative Jacobi Decoding (SJD) [6], improves Jacobi decoding with a probabilistic criterion tailored for accelerating text-to-image generation using discrete tokens. The core of SJD is a verification-refinement process. Specifically, given a sequence of tokens, in each *Jacobi iteration*, SJD first predicts the probability for each input token, and then these probabilities enable the criterion to *determine the acceptance* of a prefix of the tokens (*i.e.*, verification) while also guiding the *resampling* of unaccepted tokens for the next iteration (*i.e.*, refinement). The above verification-refinement process operates within a fixed-length sliding *Jacobi window* where the *accepted* tokens are removed and newly initialized tokens are appended. By accepting multiple tokens (at least one token) per iteration, SJD reduces model forward passes, speeding up generation compared to token-by-token decoding.

---

[*]Corresponding Author

39th Conference on Neural Information Processing Systems (NeurIPS 2025).

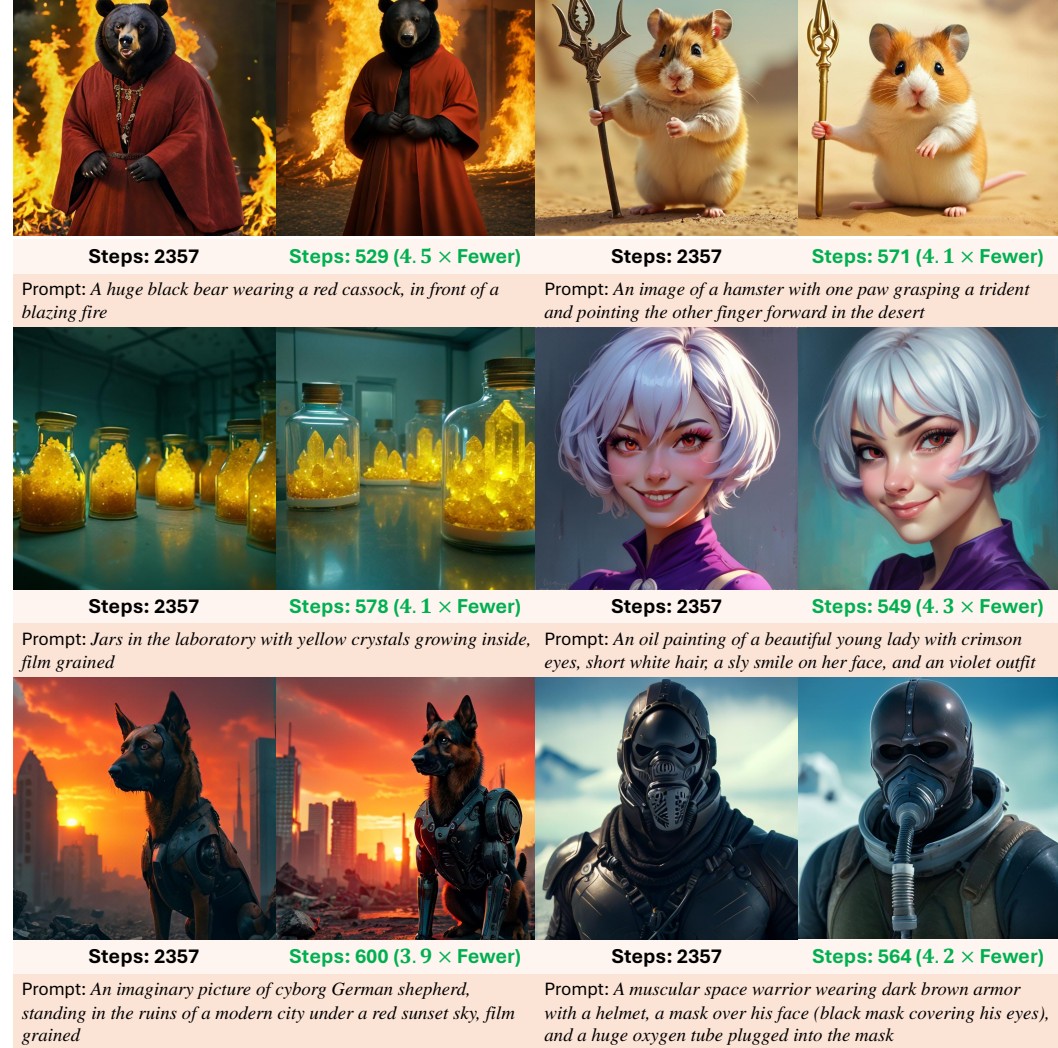

**Steps: 2357**  **Steps: 529 (4.5 × Fewer)**  **Steps: 2357**  **Steps: 571 (4.1 × Fewer)**

Prompt: *A huge black bear wearing a red cassock, in front of a blazing fire*  Prompt: *An image of a hamster with one paw grasping a trident and pointing the other finger forward in the desert*

**Steps: 2357**  **Steps: 578 (4.1 × Fewer)**  **Steps: 2357**  **Steps: 549 (4.3 × Fewer)**

Prompt: *Jars in the laboratory with yellow crystals growing inside, film grained*  Prompt: *An oil painting of a beautiful young lady with crimson eyes, short white hair, a sly smile on her face, and an violet outfit*

**Steps: 2357**  **Steps: 600 (3.9 × Fewer)**  **Steps: 2357**  **Steps: 564 (4.2 × Fewer)**

Prompt: *An imaginary picture of cyborg German shepherd, standing in the ruins of a modern city under a red sunset sky, film grained*  Prompt: *A muscular space warrior wearing dark brown armor with a helmet, a mask over his face (black mask covering his eyes), and a huge oxygen tube plugged into the mask*

Figure 1: We propose Speculative Jacobi-Denoising Decoding to accelerate autoregressive text-to-image generation via multi-token prediction. On Lumina-mGPT, the number of model forward passes for inference (denoted as *steps*) is reduced. The inference step for our decoding is marked in **green**.

Although SJD accelerates autoregressive text-to-image generation by a non-negligible margin, the token refinement process in SJD is inherently unconstrained, making it difficult to control the refinement to achieve the correct token predictions. Consequently, some tokens undergo many refinement iterations before being accepted, resulting in a low speedup ratio. In contrast, diffusion models explicitly define a trajectory for the iterative input refinement, known as the denoising process, which is governed by the principles of stochastic differential equations [7]. More importantly, previous works [8–10] have demonstrated that this trajectory can be remarkably short (as few as tens of iterations) for any length of inputs (*i.e.*, various image resolutions). Inspired by this intuition, to further accelerate the autoregressive text-to-image generation, we leverage the denoising process from the diffusion models to assist the Jacobi decoding and fine-tune the autoregressive model to adapt to the noise perturbation.

In this paper, we propose Speculative Jacobi-Denoising Decoding (SJD2), along with a fine-tuning strategy, to enable pre-trained autoregressive text-to-image generation models to perform parallel *token-level denoising* decoding. To achieve this, we introduce a task called *next-clean-token prediction*, where an autoregressive model accepts noisy input tokens and predicts the clean tokens at a one-position offset (*i.e.*, noise-free next tokens). Our noise-augmented fine-tuning strategy

equips pre-trained autoregressive models with the denoising ability, *i.e.*, we randomly select segments of *input* tokens and add Gaussian noise to their token embeddings during training. The training supervision remains consistent with conventional autoregressive models, *i.e.*, the one position-offset discrete tokens are taken as ground-truth supervision and the model is trained with cross-entropy loss. After fine-tuning, the model has the ability to process noisy inputs and thus perform *Jacobi-Denoising decoding* during inference. The decoding process begins with a sequence (Jacobi window) of token embeddings randomly initialized with Gaussian noise. In each Jacobi-Denoising iteration, this sequence is fed into the autoregressive model to predict the probability distribution of clean tokens at each token position. The noisy tokens are refined for the next iteration through the denoising process using the probability predicted in the current iteration. If a token is sufficiently denoised, it becomes a clean token and undergoes the standard Jacobi refinement. The iterative process is repeated until all tokens are accepted.

To validate our method, we conduct experiments on two open-source large autoregressive models, Lumina-mGPT [11] and Emu3 [4]. Experimental results demonstrate that our method reduces the number of forward passes by about $4\times$ on Lumina-mGPT and more than $5\times$ on Emu3 and thus achieves latency speedup by more than $2\times$. We further verify image quality to show that our method accelerates autoregressive text-to-image generation without compromising image quality.

## 2 Related Work

**Integration of Autoregression and Continuous Diffusion**   The AR-diffusion language model [12] integrates the concept of autoregression into an embedding diffusion language model [13–18]. The embedding diffusion performs the standard denoising process on (normalized) token embeddings instead of next-token prediction, and use the regression loss [12] or cross-entropy loss [14] for training. The recent diffusion-forcing model and its variants [19–25] share a similar pipeline as the AR-diffusion model and introduce the autoregressive sampling into the temporal dimension of the continuous latent video diffusion models. They employ causal masks to support history-conditioned video generation, but have independent noise levels across input tokens. Moreover, for the diffusion process on the temporal dimension, works like the rolling diffusion model [26] further introduce the sliding window mechanism on the time series. Transfusion and its variants [27–29] utilize a single backbone model to jointly perform diffusion-based image generation in a continuous latent space and autoregressive language generation in a discrete space, using separate loss, different lightweight decoders, and specific attention masks for training. MAR and its variants [30–32] propose the use of an efficient MLP diffusion head to decode features from autoregressive backbones, enabling autoregression in continuous latent space. Several multimodal large language models (MLLMs) [33–38] generate images via pre-trained diffusion models such as SDXL [39] which take the output features of autoregressive models as conditions.

**Parallel Token Decoding in Autoregressive Models**   Blockwise Parallel Decoding [40] and Medusa [41] employ auxiliary modules to predict multiple next tokens simultaneously, thereby accelerating language models. Speculative decoding [42, 43, 41, 44–47] enhances inference efficiency by using a smaller model to generate candidate tokens, which are then verified and accepted by the larger model in parallel. DiffuLLaMA [48] fine-tunes the large autoregressive language models into discrete diffusion models [49] for parallel decoding. Jacobi Decoding [5], initially applied to pixel-level autoregressive generation models, iteratively decodes tokens in parallel until their values converge, often in a training-free manner. Lookahead Decoding [50] employs the token trajectories of Jacobi decoding to form a pool of $n$-grams generated using greedy sampling to accelerate language models. CLLMs [51] collect the Jacobi trajectories into a dataset and then distill the language models with it. Speculative Jacobi Decoding [6] revisits the original Jacobi Decoding in image generation and adapts it for modern autoregressive text-to-image generation based on discrete tokens by simply introducing a probabilistic criterion. Spatially parallel image autoregressive decoding [52, 53] decodes multiple tokens simultaneously by leveraging spatial dependencies. Distilled-Decoding [17] distills an autoregressive model into a consistency model [54] via an embedding-prediction head with millions of prepared noise-token pairs and hundreds of training epochs.

**Discussion**   In contrast to the above works, this paper integrates the *denoising* process into *discrete autoregressive* models while preserving the properties like next-token prediction. Our method enables flexible modulation between the autoregression, the speculative decoding, and the denoising process.

We *fine-tune* pre-trained autoregressive text-to-image generation models to achieve our goal with a few epochs and off-the-shelf image data while avoiding adding additional modules.

## 3    Preliminaries

**Autoregressive Generation**    Let $\{x_1, \ldots, x_N\}$ denote a sequence of discrete tokens, where $N$ is the sequence length and $x_i \in V$ is an integer from a vocabulary of size $|V|$. In autoregressive models, the joint probability of the sequence is factorized as: $\mathcal{P}(x_1, x_2, \ldots, x_N) = \mathcal{P}(x_1) \prod_{i=2}^{N} \mathcal{P}(x_i | x_1, \ldots, x_{i-1})$, which assumes each discrete token $x_i$ depends only on its preceding tokens, following a causal structure. Autoregressive models parameterize the conditional probability as $\mathcal{P}_\theta(x_i | x_1, \ldots, x_{i-1})$ and sequentially *decode* (*i.e.*, sampling a token based on the predicted probability) each discrete token $x_i$ based on the preceding outputs.

**Jacobi Decoding and Speculative Jacobi Decoding**    Jacobi Decoding treats the autoregressive decoding as solving a non-linear equation in a triangular system via fixed-point iteration [5]. Instead of sequentially generating tokens by the rule $x_i \sim \mathcal{P}_\theta(x | x_1, \cdots, x_{i-1})$, Jacobi Decoding introduces an iteration index $j$ to enable parallel updates across all the sequential positions: $x_i^{(j+1)} \sim \mathcal{P}_\theta(x | x_1^{(j)}, \cdots, x_{i-1}^{(j)})$ where $x_i^{(j)}$ denotes a token at position $i$ at iteration $j$. Jacobi Decoding starts with randomly initialized tokens and iterates across the dimension $j$ until convergence, *i.e.*, the tokens remain unchanged between two consecutive iterations. Since the number of iterations is proven to be not greater than the number of tokens [5], the acceleration can be achieved.

To adapt Jacobi Decoding for the modern autoregressive text-to-image generation which is based on a large range of discrete tokens, Speculative Jacobi Decoding (SJD) [6] improves it by introducing the probabilistic criterion from speculative sampling [42, 43] to determine the convergence of tokens:

$$x_i^{(j)} \text{ converged if } r < \min\left(1, \frac{\mathcal{P}_\theta(x_i^{(j)} | x_1^{(j)}, \cdots, x_{i-1}^{(j)})}{\mathcal{P}_\theta(x_i^{(j)} | x_1^{(j')}, \cdots, x_{i-1}^{(j')})}\right), \ r \sim \mathcal{U}[0, 1], \tag{1}$$

where $\mathcal{U}[0, 1]$ denotes a uniform distribution between 0 and 1, and $j'$ is set to $j - 1$ by default, *i.e.*, the tokens generated in previous Jacobi iterations serve as draft tokens for the current Jacobi iteration. SJD also maintains a fixed-length *Jacobi window* within which Jacobi iterations are performed and token convergence is determined. In each iteration, a prefix of the tokens is determined to be converged and removed from the window (*i.e.*, the tokens are accepted), while the remaining tokens are resampled and newly initialized tokens are appended to the window.

**Continuous Diffusion Models**    Continuous diffusion models [55–59, 8, 60] generate data by learning to reverse a noise-corruption process. In this process, a clean continuous input $x_0$ is gradually corrupted into pure Gaussian noise $\epsilon$. This can be formulated as: at any timestep $t$ (normalized to the range of $[0, 1]$) the noise perturbation can be written as $x_t = \alpha_t x_0 + \sigma_t \epsilon$ where $\alpha_t$ and $\sigma_t$ are manually defined functions. As $t$ increases, $\alpha_t$ monotonically decreases but $\sigma_t$ increases. The reverse denoising trajectory between two timesteps can be solved by: $x_t = \frac{\sigma_t}{\sigma_s} x_s + \alpha_s \left(\frac{\alpha_t}{\alpha_s} - \frac{\sigma_t}{\sigma_s}\right) D_\theta(x_s, s)$ [61]. Here, $D_\theta$ is a neural network trained via regression loss to predict $x_0$ given a noisy input and a function of timestep. In this paper, we incorporate the denoising process into the Jacobi decoding.

## 4    Method

In Jacobi Decoding, the refinement of tokens is unconstrained and difficult to control, with no guarantee that tokens will follow the fastest path to reach the correct values. In contrast, diffusion models have been demonstrated to generate high-resolution, high-quality images through short trajectories (as few as dozens of iterations) [8–10]. Motivated by this, we propose integrating the denoising process from the diffusion models into Jacobi Decoding.

**Overview**    Our decoding process, illustrated in Figure 2, comprises the following steps: (1) **Token initialization:** Given a sequence of noisy normalized token embeddings (illustrated as blue-bordered patches in the first row) and prefilling/already-accepted tokens (depicted as green circles in the first row), the noise levels of the token embeddings are configured to be non-strictly monotonically

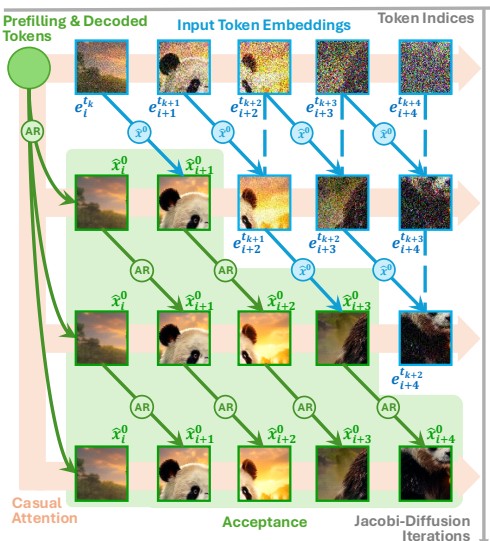

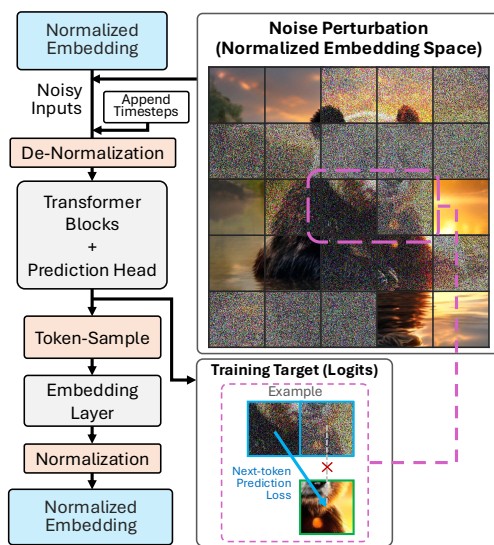

Figure 2: Overview of our decoding process. The noisy token embeddings with increasing noise levels undergo a parallel forward pass with a causal attention mask, predicting conditional probabilities and then sampling clean tokens. A probabilistic criterion selects a prefix of tokens for acceptance (green area). For unaccepted tokens, if clean, the next-token prediction performs on them (green solid arrows marked with AR). If noisy, they are denoised with one-position offset (blue solid $\hat{x}^0$ arrows and dash arrows).

Figure 3: Overview of our training strategy and model process. The input token embedding sequence is randomly divided into segments and the adjacent segments are perturbed with the noise of consecutive levels. The noisy embeddings with timestep tokens are fed into transformer blocks and a prediction head. During training, the predicted probability is used to compute the cross-entropy loss for next-token prediction. During inference, the probability is for token sampling and then generating embeddings.

increasing. Before the iterations start, the sequence is initialized with pure Gaussian noise. (2) **Parallel forward:** After initialization, these token embeddings undergo a Jacobi-denoising iteration, where they are fed into the neural network along with timestep encodings for a single parallel forward pass using a causal attention mask. The network predicts conditional probability and performs token sampling for the next clean token at each position. Sampled tokens from noisy embedding inputs are denoted as the one-position-offset $\hat{x}^0$-predictions (marked by the down-right blue solid arrows) while those from prefilling or accepted token inputs are denoted as autoregressive (AR) predictions, *i.e.*, the standard next-token prediction (marked by green solid arrows). (3) **Verification:** After the parallel forward pass, a prefix of sampled tokens is accepted based on the probabilistic criterion outlined in Equation (1). For example, in the second row of Figure 2, the first two sampled tokens are accepted and marked with green borders. (4) **Refinement for unaccepted tokens:** After the verification, the refinement is performed for unaccepted tokens. For each unaccepted noisy token, a denoising step, outlined in Equation (3), is performed on its embedding. Specifically, a linear combination is performed on the token embedding from the previous iteration at the same spatial position (indicated by vertical blue dashed lines) and the embedding of the predicted clean tokens from the one-position offset (the down-right blue solid arrows). If an unaccepted token has been sufficiently denoised to be clean in previous iterations, there is no further denoising needed and the refinement follows the standard next-token prediction. This iterative process repeats until all required tokens are accepted, serving as the final outputs.

## 4.1 Model Parameterization

To seamlessly integrate the denoising process with next-token prediction, we propose a paradigm of *next-clean-token prediction* based on *noisy normalized token embeddings*. In this paradigm, the autoregressive neural network learns to accept noisy input tokens and predict the clean tokens at a one-position offset (*i.e.*, noise-free next tokens). Let $e$ denote a normalized token embedding and $\theta$

denote the autoregressive model, our paradigm can be formulated as follows:

$$\hat{x}_{i+1}^0 \sim \mathcal{P}_\theta\big(x|\boldsymbol{e}_1^0, \cdots, \boldsymbol{e}_{i'-1}^0, \boldsymbol{e}_{i'}^{t_0}, \cdots, \boldsymbol{e}_i^{t_k}\big),$$
$$\hat{\boldsymbol{m}}_{i+1}^0 = \boldsymbol{W}_e \cdot \text{One-hot}(\hat{x}_{i+1}^0), \tag{2}$$
$$\hat{\boldsymbol{e}}_{i+1}^0 = \text{Normalize}(\hat{\boldsymbol{m}}_{i+1}^0),$$

where $\boldsymbol{e}_i^t$ is the normalized token embedding at spatial position $i$ in the sequence and at timestep $t$, and $\hat{\boldsymbol{e}}_{i+1}^0$ represents the predicted clean normalized token embedding at position $i + 1$, $\mathcal{P}_\theta\big(x|\boldsymbol{e}_1^0, \cdots, \boldsymbol{e}_{i'-1}^0, \boldsymbol{e}_{i'}^{t_0}, \cdots, \boldsymbol{e}_i^{t_k}\big)$ denotes the predicted conditional probability of a discrete token $x$ with the embedding sequence $\{\boldsymbol{e}_1^0, \cdots, \boldsymbol{e}_{i'-1}^0, \boldsymbol{e}_{i'}^{t_0}, \cdots, \boldsymbol{e}_i^{t_k}\}$ as conditions, One-hot$(\cdot)$ denotes transforming a token category into a one-hot vector, $\boldsymbol{W}_e \in \mathbb{R}^{D \times |V|}$ denotes the learned embedding weight matrix of the model, and $\hat{\boldsymbol{m}}_{i+1}^0 \in \mathbb{R}^D$ denotes the $D$-dimensional predicted token embedding at position $i + 1$. Normalize$(\cdot)$ is normalizing embeddings with their statistics (details in Section 4.3). The timesteps $\{t_0, \cdots, t_k\}$, *i.e.*, noise levels, are configured to be non-strictly monotonically increasing, and we discuss the specific selection of timesteps for decoding in Section 4.2. In this paradigm, the pre-trained autoregressive models still predict the categorical probabilities of tokens instead of the continuous token embeddings, and we employ these probabilities to generate the embeddings.

## 4.2 Speculative Jacobi-Denoising Decoding

**Jacobi Window and token initialization**   In practice, our method employs a sliding Jacobi window during the decoding phase, rather than directly performing the decoding process at fixed token positions. The Jacobi window is a fixed-length sliding window containing noisy normalized token embeddings as the draft. Initially, the window is filled with pure Gaussian noise. In each iteration, a prefix of accepted tokens is removed, and an equal number of new tokens, sampled from pure Gaussian noise, is appended. Consequently, the noise levels of the tokens within the window are non-strictly monotonically increasing across iterations.

**Parallel Forward and Verification**   The model takes draft token embeddings as the inputs and predicts the conditional probability of their next clean tokens in parallel. Then, Equation (1) is employed to determine whether to accept or reject each draft token which is transformed from the input token embedding. This transformation is finding the nearest discrete tokens in the vocabulary through cosine similarity [47].

**Refinement with Denoising**   After verification, the unaccepted tokens will go through a refinement process to refine the token embeddings for the next iteration. The refinement of noisy tokens is realized by the denoising process. For denoising, we first define a *fixed* monotonically decreasing timestep sequence $\{t_K, \cdots, t_k \cdots, t_0, 0\}$, where $t_0$ is set very close to zero [62]. The values of these timesteps can follow the Karras timestep scheduler [62]. Then, we show the specific denoising formula based on these timesteps when $k > 0$:

$$\boldsymbol{e}_i^{t_{k-1}} = \frac{\sigma_{t_{k-1}}}{\sigma_{t_k}} \boldsymbol{e}_i^{t_k} + \alpha_{t_k}\left(\frac{\alpha_{t_{k-1}}}{\alpha_{t_k}} - \frac{\sigma_{t_{k-1}}}{\sigma_{t_k}}\right)\hat{\boldsymbol{e}}_i^0, \tag{3}$$

where $\hat{\boldsymbol{e}}_i^0$ is the token embedding predicted according to Equation (2). In the above equation, we perform a denoising step like the diffusion models to obtain the embedding with a smaller noise level at position $i$ with timestep $t_k$. If $k = 0$, *i.e.*, the denoising process is just complete, Equation (3) reduces into the standard Jacobi iteration: $\boldsymbol{e}_i^0 := \boldsymbol{0} + 1 \cdot \hat{\boldsymbol{e}}_i^0$. Additionally, for the unaccepted sufficiently denoised tokens, as no further denoising is required, we resample the tokens whose threshold from Equation (1) is below 0.5 but retain the others for the next Jacobi iteration.

By introducing the denoising trajectories into the decoding process of autoregressive models, our method stabilizes the token trajectories in the decoding process, accelerating the token convergence.

## 4.3 Fine-tuning Strategy

In this section, we introduce the strategy of fine-tuning a pretrained autoregressive text-to-image generation model to accept noisy input tokens for Speculative Jacobi-Denoising Decoding. Figure 3 illustrates an overview of our training strategy and the specific process of the neural network: During

training, the normalized embeddings (initially clean) are first transformed into noisy embeddings. For example, in the right-side image of Figure 3, noise levels increase non-monotonically across patches (*i.e.*, token positions) in a raster scan order (left to right, top to bottom). When reaching a randomly determined position, the noise level stops increasing and the noise level of the next position resets to zero, forming segments with non-monotonically increasing noise levels in the token sequence. Next, these noisy embeddings are appended with timestep encodings, which indicate the noise level at each position. Together, the embeddings and encodings are denormalized and are fed into transformer blocks and a prediction head to produce logits for each position. The cross-entropy loss is then applied to each position, using the clean token indices as labels, with one-position offset (shown by the dotted frame in Figure 3) for next-clean-token prediction. During inference, the input normalized embeddings are already noisy and are not further perturbed. These embeddings are also appended with timestep encodings and processed to generate logits through the denormalization, the transformer blocks and the prediction head. As described in Equation (2), these logits are used for token sampling, and these sampled clean tokens are then transformed into normalized token embeddings.

**Noise Perturbation**  We add noise to the embeddings of these discrete input tokens, bypassing the discrete values. Although we can directly perform a linear combination of the embedding and a Gaussian random variable, the distribution of the *pre-trained* embeddings may deviate from the scale of the standard Gaussian distribution. For instance, if the variance of the embedding is small, the *pre-trained* transformer blocks may only handle small values, making the values from a standard Gaussian distribution unsuitable for these blocks. To address this issue, we add noise to the *normalized* token embeddings. Subsequently, these noisy normalized embeddings are de-normalized and then fed into the transformer blocks. The detailed procedure is as follows:

$$
\begin{aligned}
\boldsymbol{e}^* &= \text{Normalize}(\boldsymbol{m}^*) = \frac{1}{\boldsymbol{\sigma}_e} \odot (\boldsymbol{m}^* - \boldsymbol{\mu}_e), \\
\boldsymbol{e}^t &= \alpha_t \boldsymbol{e}^* + \sigma_t \boldsymbol{\epsilon}, \\
\boldsymbol{m}^t &= \text{De-normalize}(\boldsymbol{e}^t) = \boldsymbol{\sigma}_e \odot \boldsymbol{e}^t + \boldsymbol{\mu}_e,
\end{aligned}
\tag{4}
$$

where $\boldsymbol{m}^* \in \mathbb{R}^D$ denotes the $D$-dimensional embedding of a ground-truth token. $\boldsymbol{\mu}_e \in \mathbb{R}^D$ and $\boldsymbol{\sigma}_e \in \mathbb{R}^D$ denote the mean and standard deviation of the learned embedding weight $\boldsymbol{W}_e \in \mathbb{R}^{|V| \times D}$, respectively. In practice, we directly average this weight across its first dimension to compute the mean, and the standard deviation is also obtained across this dimension. $\frac{1}{\boldsymbol{\sigma}_e} \in \mathbb{R}^D$ represents the element-wise reciprocal of $\boldsymbol{\sigma}_e$, and $\odot$ is the element-wise product. $\alpha_t \in \mathbb{R}$ and $\sigma_t \in \mathbb{R}$ are the hyper-parameters for the denoising timestep $t$, and $\boldsymbol{\epsilon} \in \mathbb{R}^D$ denotes the standard Gaussian noise. With Equation (4), we can transform a clean embedding $\boldsymbol{m}^*$ into a noisy embedding $\boldsymbol{m}^t$. For fine-tuning, the input sequence is divided into randomly sized segments, and we add the identical level of noise to the tokens from the same segment.

**Finetuning Objective and Loss Function**  We fine-tune a pre-trained autoregressive text-to-image model to predict the next clean token from the inputs of noisy token embeddings. The model accepts noisy token embeddings and outputs logits representing the categorical probability distribution of the next clean token. The cross-entropy loss is computed between these logits and the ground truth token categories, optimizing the model to denoise inputs while maintaining autoregressive prediction. Specifically, analogous to standard next-token prediction, the cross-entropy loss is performed as follows: $\mathcal{L} = \sum_{i=0}^{N} \text{Cross-Entropy}(\boldsymbol{x}_{i+1}^*, \hat{\boldsymbol{p}}_{i+1}^0)$, where $\boldsymbol{x}_i^*$ denotes the one-hot label of the ground-truth token at position $i$, $N$ is the total number of tokens, and $\hat{\boldsymbol{p}}_{i+1}^0$ denotes the conditional probability in Equation (2). Here, we assume that only the first token is prefilled with text conditioning. With this training objective and parameterization, the model learns to decode or verify clean tokens even when noisy tokens are present in the input conditions.

**Timestep Injection**  Injecting the information of timesteps into noisy inputs is a common design for the denoising process [56, 63, 64]. To avoid introducing additional modules like AdaLN [56], we take the sinusoidal encodings of timesteps as a sequence of special token embeddings and append them to the sequence of input token embeddings during fine-tuning and decoding. Then, the sequence, which comprises input token embeddings and timestep encodings, is fed into the transformer blocks. Within the attention modules of these blocks, we use the attention mask to force each noisy token embedding to attend to the corresponding timestep encoding which indicates its noise level. To ensure

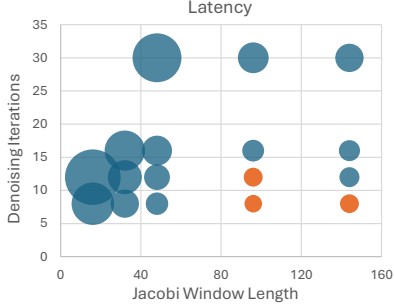

Figure 4: Correlation between denoising iterations and Jacobi window length on the latency. Circle areas represent absolute latency values, and the lowest latency (a difference within 3 seconds is allowed) is in orange.

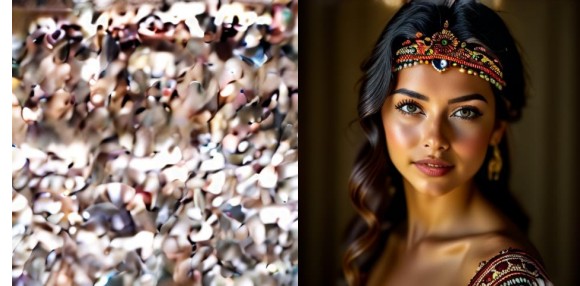

Prompt: *Miss Mexico portrait of the most beautiful mexican woman, Exquisite detail*

Figure 5: Study on embedding normalization for denoising process. Left: Denoising output without embedding normalization, failing to generate a coherent image. Right: Denoising output with embedding normalization, generating a semantically meaningful image.

Table 1: Evaluation on the validation sets of MSCOCO [65].

| Configuration | COCO2017 (5k) | | | | COCO2014 (30k) | | | |
|---|---|---|---|---|---|---|---|---|
| | Average Steps (↓) | Step Compression (↑) | FID | CLIP-Score | Average Steps (↓) | Step Compression (↑) | FID | CLIP-Score |
| Lumina-mGPT [11] | 2357 | 1.00× | 30.8 | 31.3 | 2357 | 1.00× | 21.0 | 31.3 |
| SJD [6] | 1060 | 2.23× | 31.1 | 31.3 | 1057 | 2.23× | 20.8 | 31.3 |
| **SJD2** | **592** | **4.02×** | 31.4 | 31.8 | **599** | **3.93×** | 21.1 | 31.9 |
| Emu3 [4] | 8193 | 1.00× | 31.1 | 31.0 | 8193 | 1.00× | 19.3 | 31.2 |
| SJD [6] | 3528 | 2.32× | 30.6 | 30.9 | 3535 | 2.32× | 21.4 | 31.0 |
| **SJD2** | **1461** | **5.62×** | 31.5 | 30.4 | **1461** | **5.63×** | 21.8 | 30.8 |

that the distribution of the timestep encodings aligns with that of the token embeddings, we apply a normalization-then-denormalization process to these encodings similar to Equation (4).

## 5   Experiments

### 5.1   Implementation Details

We perform experiments on two baselines, Lumina-mGPT [11] and Emu3 [4]. When generating an image at least $720 \times 720$, Lumina-mGPT only needs about 2k tokens while Emu3 requires more than 8k tokens because of the difference between their tokenizers. For each fine-tuning, 8 GPUs with 80G memory are required for each model. Since all model parameters are used for fine-tuning, we leverage DeepSpeed ZeRO-3 or FSDP with gradient checkpointing to save GPU memory at the cost of increased training time. The global batch size is set to 64, and the learning rate is set to $2 \times 10^{-5}$ with the AdamW optimizer. We tune each model only within 6 epochs, costing approximately $14 \times 8$ A100 hours for Lumina-mGPT and $26 \times 8$ H100 hours for Emu3. By default, we set classifier-free guidance to 3.0 and use top-2000 for the quantitative results of our method.

**Evaluation metrics.** To assess visual quality, we employ two key metrics: FID [66] and CLIP-Score [67] on COCO benchmark [65] and GenEval benchmark [68]. To quantify the efficiency of the decoding process, we introduce the *step compression ratio* [50]: $\mathcal{S} = \frac{\text{\# generated tokens}}{\text{\# decoding steps}}$, which serves as a theoretical measure of parallelization and thus reflects the acceleration. For each benchmark, we compute the average step compression ratio across all generated images. Additionally, this ratio is included alongside individual image samples in qualitative comparisons to highlight the performance differences between our method and other approaches. More importantly, we provide the latency of model forward passes on a single GPU to evaluate the practical speedup achieved.

Table 2: Visual quality on GenEval benchmark [68].

| Method | Colors | Position | Counting | Two | Color Attri | Single | Overall |
|---|---|---|---|---|---|---|---|
| Lumina-mGPT [11] | 0.83 | 0.09 | 0.26 | 0.60 | 0.15 | 0.96 | 0.48 |
| Lumina-mGPT + SJD [6] | 0.82 | 0.08 | 0.23 | 0.63 | 0.12 | 0.96 | 0.47 |
| Lumina-mGPT (tuned) | 0.81 | 0.13 | 0.27 | 0.65 | 0.27 | 0.98 | 0.52 |
| Lumina-mGPT (tuned) + SJD [6] | 0.81 | 0.12 | 0.31 | 0.68 | 0.24 | 0.97 | 0.52 |
| Lumina-mGPT (tuned) + **SJD2** | 0.79 | 0.11 | 0.31 | 0.64 | 0.23 | 0.96 | 0.51 |
| Emu3 [4] | 0.78 | 0.15 | 0.33 | 0.69 | 0.16 | 0.98 | 0.52 |
| Emu3 + SJD [6] | 0.79 | 0.12 | 0.28 | 0.61 | 0.13 | 0.97 | 0.48 |
| Emu3 (tuned) + **SJD2** | 0.73 | 0.14 | 0.28 | 0.61 | 0.24 | 0.96 | 0.49 |

## 5.2 Qualitative Results

In Figure 1 and Figure 6, we compare the standard autoregressive decoding and our method on Lumina-mGPT [11]. The results show that our method can achieve about $4\times$ fewer steps for autoregressive text-to-image generation and the visual quality is preserved. We also compare different decoding methods on Emu3 [4] in Figure 7.

## 5.3 Quantitative Results

Our SJD2 significantly reduces the steps for autoregressive text-to-image generation while maintaining visual quality, as demonstrated by our evaluation on MS-COCO [65] validation sets in Table 1. We also find that our method achieves a higher step compression ratio on Emu3 [4] than that on Lumina-mGPT [11]. Specifically, SJD2 can achieve a step compression about $4.0\times$ on Lumina-mGPT and about $5.6\times$ on Emu3. For visual quality, we further compare our method to autoregressive decoding and SJD [6] on the GenEval benchmark [68] with Lumina-mGPT [11] as the baseline in Table 2. Our method achieves an overall score of $0.51$, nearly matching the $0.52$ of tuned AR and SJD, demonstrating preserved visual quality. More importantly, following [47], we select 100 COCO prompts on the identical A100 server to compare the practical average speedup and visual quality among these decoding methods. According to the latency reported in Table 3, our method is still faster than other decoding methods on the real server by more than $2\times$. We also provide the GPU memory usage during inference. While our parallel decoding method achieves significant latency reductions, we acknowledge it incurs an additional memory overhead of about 3GB compared to autoregressive decoding, because of the variables for denoising like the timestep tokens.

## 5.4 Ablation Studies

We perform experiments for our method with Lumina-mGPT as the baseline and on one RTX 4090 by default. The selected 100 prompts used in Table 3 are for the evaluation in the ablation studies. We also include a specific and detailed analysis for our denoising process in Appendix B.

**Study on the sampling timesteps and Jacobi window length.** While contemporary diffusion models typically require dozens of denoising iterations to achieve satisfactory results, extending Jacobi window lengths introduces computational overhead. This establishes a trade-off between denoising iteration counts and Jacobi window length for the minimization of latency. According to the results in Figure 4, when constraining denoising steps to 20 while maintaining Jacobi window lengths beyond 80, the latency reduction converges to a minimum.

**Study on the embedding normalization.** When verifying the usefulness of embedding normalization, we focus on our denoising process by enforcing the denoised tokens to be immediately accepted (as detailed in Appendix B). As shown in Figure 5, deactivating embedding normalization results in a complete failure of the denoising process, yielding pure noise instead of coherent images. Conversely, activating normalization enables the denoising process to generate reasonable outputs, demonstrating the critical role of embedding normalization in our denoising process.

Table 3: The computational cost of decoding methods on a subset of COCO prompts.

| Methods | Steps | Latency | CLIP-Score | GPU Memory |
|---------|-------|---------|------------|------------|
| Lumina-mGPT [11] | 2357 | 88.55s | 32.0 | 17G |
| SJD [6] | 1058 | 41.99s | 31.5 | 17G |
| **SJD2** | **596** | **33.64**s | 32.2 | 20G |
| Emu3 [4] | 8193 | 375.29s | 30.9 | 20G |
| SJD [6] | 3537 | 207.60s | 31.2 | 20G |
| **SJD2** | **1470** | **147.65**s | 30.7 | 23G |

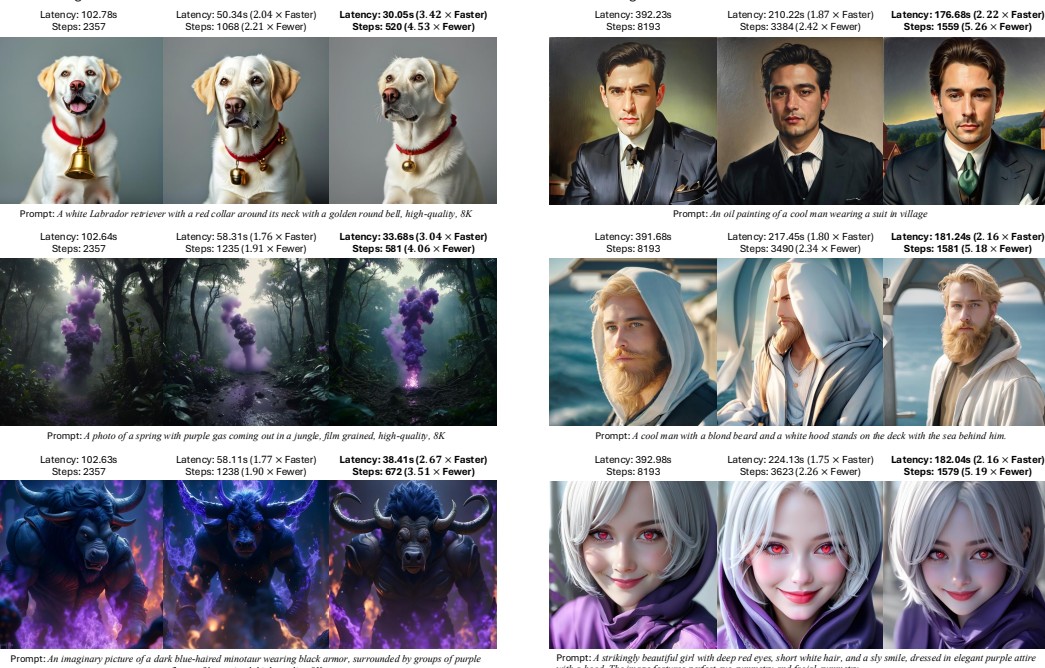

Figure 6: The comparison of the original autoregressive decoding, SJD [6], and our method with Lumina-mGPT [11] as the baseline and on one RTX 4090.

Figure 7: The comparison of the original autoregressive decoding, SJD [6], and our method with Emu3 [4] as the baseline and on one RTX 4090.

## 6 Conclusion

This paper introduces Speculative Jacobi-Denoising Decoding, a new algorithm integrating the continuous denoising process into Jacobi decoding to accelerate autoregressive text-to-image generation. By extending next-token prediction into next-clean-token prediction with noisy inputs, we enable pre-trained autoregressive models to learn to denoise noise-perturbed token embeddings through a fine-tuning strategy. The proposed Jacobi-Denoising decoding initializes token sequences with Gaussian noise. It then iteratively refines them using a process that combines denoising steps with Jacobi decoding and an improved probabilistic prefix acceptance criterion. Experiments show that our method can reduce the number of model forward passes for acceleration while keeping the visual quality of generated images.

## Acknowledgment

This work is supported by the National Nature Science Foundation of China (No. 62402406).

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

# Appendix

## A  More Implementation Details

**Experimental settings.** We perform experiments on two baselines, Lumina-mGPT [11] and Emu3 [4]. We have collected approximately 80k high-resolution (at least $720 \times 720$) images from the Internet. For images lacking text descriptions, we caption them with Qwen2-VL [69]. We follow the advanced flow matching setting [70, 71] with $\alpha_t = 1 - t$ and $\sigma_t = t$ for our denoising process. By default, the length of the Jacobi window of our method is set to 96 for Lumina-mGPT and 128 for Emu3. The number of denoising steps in SJD2 is set to 25.

## B  Further Analysis

**The pure denoising process also generates reasonable images.** In our approach, the speculative Jacobi decoding is applied after the denoising steps, so we need to recognize that the acceleration benefits stem not only from the Jacobi iterations but also from the effectiveness of the denoising process. To systematically verify that the denoising stage alone can produce valid tokens, we conduct the following experiment: when a token is successfully denoised and its distance from the first point of the Jacobi window is smaller than $\frac{L}{T}$ (where $L$ represents the length of the Jacobi window and $T$ denotes the number of denoising iterations), we immediately accept it. The ratio $\frac{L}{T}$ guarantees that the last token in the Jacobi window completes exactly $T$ denoising iterations when it reaches the left part of the window. The results of this experiment are presented in Figure 9a. According to these results, we observe that our denoising process can generate reasonable images in certain cases, particularly those featuring characters. Additionally, the immediate acceptance of denoised tokens significantly reduces the number of steps and overall latency. In comparison, as shown in Figure 9b, we reintroduce Jacobi iterations into the decoding process. These iterations serve as a refinement mechanism, enhancing the image generation with more intricate details and reducing the artifacts. However, this improvement comes at the cost of increased computational steps. Therefore, to preserve image quality, we opt to keep the Jacobi iterations in our method.

**Analysis of unifying noise perturbation for discrete and continuous inputs.** We demonstrate the feasibility of unifying noise perturbation for both the discrete and continuous inputs in the state-of-the-art transformer-based models. Since the noise perturbation is commonly used in diffusion models, we first analyze the behavior of noisy inputs in diffusion transformer (DiT) architecture. Although noise perturbation appears to occur in the latent space, the learnable linear transformation $\boldsymbol{W}$ : $\mathbb{R}^{dHW} \to \mathbb{R}^{DH_SW_S}$ (usually implemented by a 2D convolution without receptive field overlapping) before the transformer blocks causes the perturbation to actually happen in the feature space: $\boldsymbol{W}\boldsymbol{x}_t = \alpha_t(\boldsymbol{W}\boldsymbol{x}^*) + \sigma_t\boldsymbol{n}$, where $\boldsymbol{n} \in \mathbb{R}^{DH_SW_S}$ is a random Gaussian variable formed by a linear weighted sum of independent Gaussian noises with the elements of $\boldsymbol{W}$ as weights, and $D, H_S, W_S, d$ denote feature dimension, latent height, latent width and latent dimension, respectively. Therefore, based on the above equation, the noise perturbation can be interpreted as a linear combination of a clean feature vector and a noise vector. Since both DiTs and autoregressive models rely on transformer blocks operating in the feature space, we aim to align their noise perturbation on the input embedding.

**Analysis of Flops in inference.** We present the average Flops per output token in Table 4. The results reveal that autoregressive decoding requires fewer Flops than SJD2. Although more Flops are used for decoding, the practical latency becomes lower. Actually, this Flops overload stems from the paradigm of all the speculative decoding methods. Their drafting-and-verification mechanism, which inherently introduces computational overhead: the number of accepted tokens per sampling step is substantially lower than the number of input draft tokens.

**Smaller baseline.** We also implement SJD2 on Janus-pro-1B [72], an advanced autoregressive model much smaller than Lumina-mGPT [11]. The results are in Table 5 and Table 6. From the results, we observe that SJD2 still can accelerate Janus-pro without sacrificing on generated image quality, as evidenced by the following Geneval metrics [68].

**Comparison to other accelerating methods.** We compare our method with the recent and classic speculative/parallel decoding methods, including Lantern [47], ZipAR [53], Eagle [46] and Jacobi

Table 4: The inference Flops of autoregressive models with different decoding methods.

| Method | GFlops (↓) | Latency (↓) |
|---|---|---|
| Lumina-mGPT [11] | 18.72 | 88.55s |
| Lumina-mGPT + **SJD2** | 219.60 | 33.64s |
| Emu3 [4] | 18.15 | 375.29s |
| Emu3 + **SJD2** | 465.92 | 147.65s |

Table 5: Inference performance comparison on Janus-Pro-1B [72].

| Method | Latency (↓) | Steps (↓) |
|---|---|---|
| Janus-Pro-1B [72] | 9.1s | 576 |
| Janus-Pro-1B + **SJD2** | 2.5s | 144 |

Decoding [5], on COCO2017 validation set [65] with Lumina-mGPT [11] as baseline. As shown in Table 7, our approach achieves superior acceleration while maintaining comparable visual quality.

**Comparison to diffusion models.** While many autoregressive models currently underperform state-of-the-art diffusion models in image quality and face acceleration challenges, our SJD2 narrows the speed gap. In Table 8, we evaluate inference latency for several commonly-used diffusion models (smaller than 3B) and Janus-Pro [72] at the same resolution ($384 \times 384$). We set the number of sampling steps for diffusion models as 50. Results demonstrate that our SJD2 reduces latency of Janus-Pro-1B from 9.1s to 2.5s, narrowing the gap between Janus-pro and the advanced diffusion models like SD3. Moreover, this result means Janus-Pro-1B with SJD2 already outperforms SDXL in speed (2.5s vs. 4.3s).

**Investigation on the refinement.** In this paragraph, we demonstrate that SJD2 stabilizes the refinement trajectory, as illustrated in Figure 8. Specifically, we apply SJD2 and SJD [6] on Emu3 [4] respectively, and then we examine the first five tokens from the Jacobi window in one iteration, computing the times of the token change between adjacent steps and the cumulative changes. As shown in the first five figures, SJD2 yields identical initial predictions across the 25 sampling steps. As the noise level decreases, token predictions diversify but then become unchanged at several steps, indicating the stabilization of the token trajectory. In contrast, for SJD, the tokens consistently change, appearing to oscillate and remain unstable. The last figure of Figure 8 also shows that the SJD causes more times of token change than SJD2.

## C    Limitations and Future Work

Although SJD2 can achieve similar step compression in various models, the improvements on actual latency are not consistent, shown by Figure 6 and Figure 7. We speculate that this is caused by the different sizes of KV cache in different autoregressive models (the model whose tokenizer has a low image compression ratio leads to a large KV cache). A promising direction is to stabilize the latency acceleration of Jacobi-based acceleration methods across the models with different KV caches.

Table 6: Visual quality on Geneval benchmark [68] with Janus-pro-1B [72] as the baseline.

| Method | Colors | Position | Counting | Two | Color Attri | Single | Overall |
|---|---|---|---|---|---|---|---|
| Janus-Pro-1B (tuned) [72] | 0.82 | 0.39 | 0.39 | 0.55 | 0.42 | 0.94 | 0.59 |
| Janus-Pro-1B (tuned) + **SJD2** | 0.83 | 0.45 | 0.37 | 0.59 | 0.38 | 0.96 | 0.60 |

Table 7: Comparison to other accelerating methods with Lumina-mGPT [11] as baseline.

| Configuration | Acceleration Latency (↑) | Acceleration Step (↑) | CLIP-Score (↑) |
|---|---|---|---|
| Autoregressive Decoding | 1.00× | 1.00× | 31.3 |
| Jacobi Decoding [5] | 1.02× | 1.04× | 31.4 |
| SJD [6] | 2.05× | 2.23× | 31.3 |
| EAGLE [46] | 2.10× | 2.94× | 33.3 |
| LANTERN [47] | 2.56× | 3.63× | 32.7 |
| ZipAR [53] | 1.82× | 4.00× | 31.2 |
| **SJD2** | **2.63×** | **4.02×** | 31.8 |

Table 8: Efficiency comparison with the diffusion model.

| Method | Latency (↓) | Steps (↓) |
|---|---|---|
| Janus-Pro-1B [72] | 9.1s | 576 |
| SD1.5 [55] | 1.7s | 50 |
| SDXL [39] | 4.3s | 50 |
| SD3-Medium [70] | 1.7s | 50 |
| Janus-Pro-1B + **SJD2** | 2.5s | 144 |

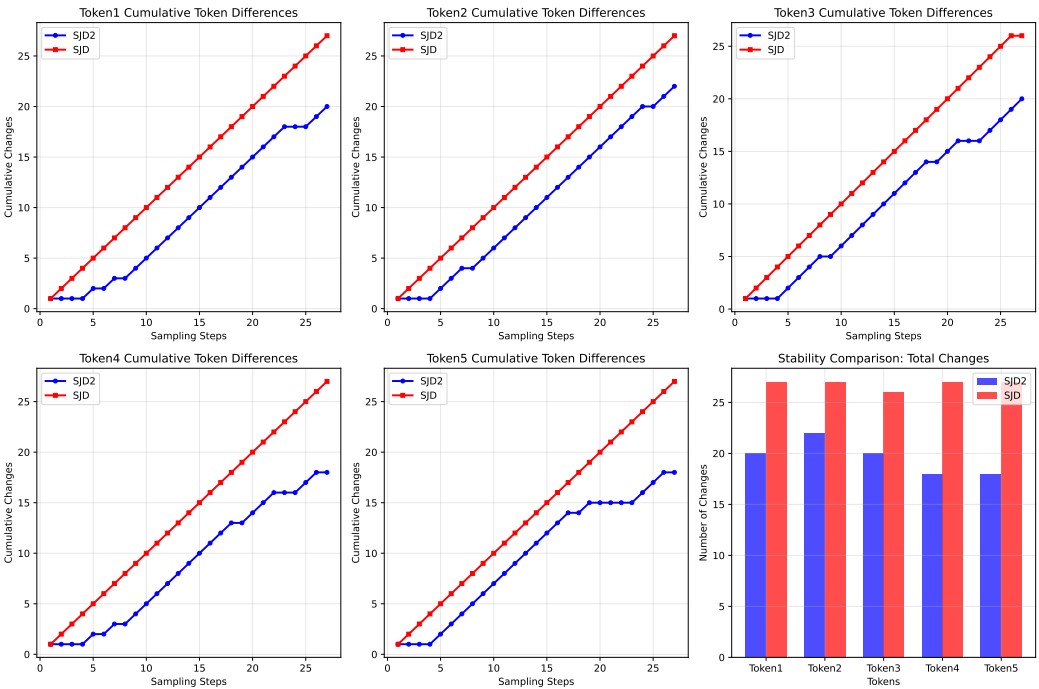

Figure 8: The trajectories of token difference.

# D   Impact Statement

Our paper proposes a new method of autoregressive text-to-image generation for research purposes. Real-world deployment would require additional safeguards beyond technical implementation. We recognize that this open technical accessibility could lead to potential societal risks, including misuse for generating misleading content or harmful biases. Fortunately, this problem can be alleviated by strict dataset filtering to exclude harmful content.

| Latency: 19.49s | Latency: 19.35s | Latency: 19.40s |
| Steps: 306 | Steps: 305 | Steps: 306 |

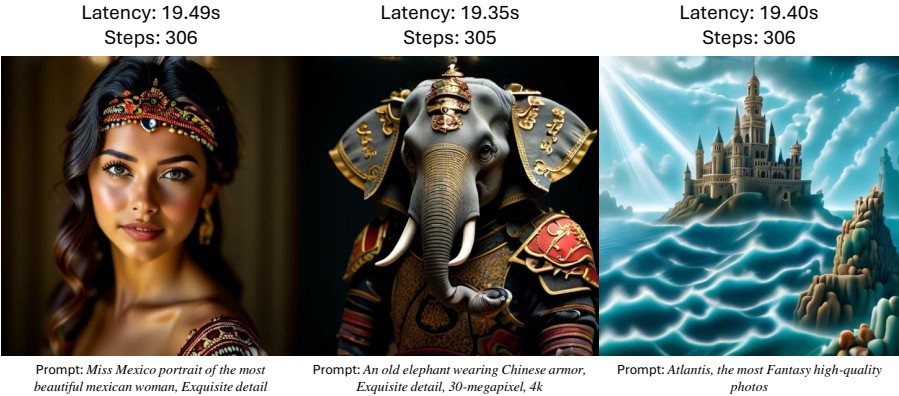

Prompt: *Miss Mexico portrait of the most beautiful mexican woman, Exquisite detail*    Prompt: *An old elephant wearing Chinese armor, Exquisite detail, 30-megapixel, 4k*    Prompt: *Atlantis, the most Fantasy high-quality photos*

(a) The images generated with the immediate acceptance of denoised tokens (without Jacobi iterations).

| Latency: 32.80s | Latency: 34.83s | Latency: 35.18s |
| Steps: 538 | Steps: 575 | Steps: 587 |

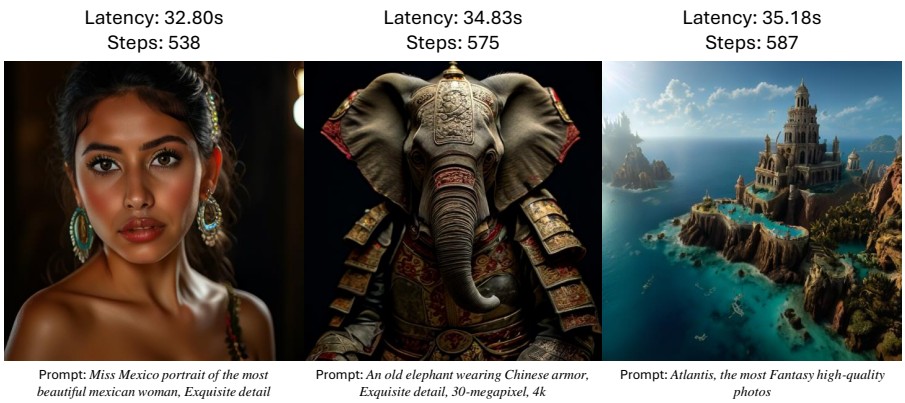

Prompt: *Miss Mexico portrait of the most beautiful mexican woman, Exquisite detail*    Prompt: *An old elephant wearing Chinese armor, Exquisite detail, 30-megapixel, 4k*    Prompt: *Atlantis, the most Fantasy high-quality photos*

(b) The images generated with the combination of Jacobi iterations and denoising.

Figure 9: Analysis of our denoising process. (a) Our denoising process without Jacobi iterations can generate reasonable images with few model forward passes and small latency. (b) The further Jacobi iterations refine the results of our denoising process, resulting in more details and fewer artifacts.

