# OpenReview forum: "Speculative Jacobi-Denoising Decoding for Accelerating Autoregressive Text-to-image Generation"
_NeurIPS.cc/2025/Conference — NeurIPS 2025 poster_

### Official Review · Reviewer_3CPQ · 2025-06-30

**Clarity:** 3
**Significance:** 3
**Originality:** 3
**Rating:** 4
**Confidence:** 4

**Summary:**

This paper proposes Speculative Jacobi-Denoising Decoding to enable the control of refinement in SJD. A novel and sophisticated fine-tuning strategy by next-clean-token prediction is designed to predict the clean token at the next position with noised tokens as input. Then a part of perfix tokens would be selected after verification and the remaining tokens would be refined. The experiments show that the proposed method largely accelerates the inference speed and maintains the generation quality.

**Questions:**

1. Is the proposed SJD2 able to perform batch inference?
2. What's the specific choice of $\alpha_t$ and $\sigma_t$ in Equation (3)?

**Ethical Concerns:**

["NO or VERY MINOR ethics concerns only"]

**Final Justification:**

The author's clever rebuttal addressed my main concerns, and I tend to remain positive rating.

**Limitations:**

Yes

**Quality:**

4

**Strengths And Weaknesses:**

Strength:

1. The proposed SJD2 is novel and shows impressive acceleration and maintains the generation quality.
2. The proposed fine-tuning strategy enables low-cost fine-tuning for the consecutive acceleration method.

Weakness:

1. The method's description is dense and requires prior familiarity with Jacobi decoding, speculative decoding, and diffusion models. This may hinder readability on readers who are not familiar with some of these topics.
2. While the denoising process is justified through analogy to diffusion models, the behavior of denoising in discrete token space is not fully analyzed. More investigations on the claim that diffusion models improve the refinement, like showing how the output tokens converge, can be helpful.

---

> ### Author Rebuttal · Authors · 2025-07-31
>
> # Response to Reviewer 3CPQ
>
> Thank you for your thoughtful and constructive feedback on our paper. We are particularly grateful for your recognition of our novelty and high performance.
>
> ## 1. **Enhanced prelimiary on speculative decoding and Jacobi decoding**
> Thank you for your suggestion. To improve clarity and readability, we improve preliminary section and add more knowledge details related to our core design. Specifically, we refine the descriptions of speculative decoding and Jacobi decoding: **(A) speculative decoding**: Speculative decoding employs a small model to accelerate the sequential generation of large autoregressive model. This model is trained on the same domain as the large model and is small enough for faster generation. In each step of inference, the small model first generates a sequence with its own inference paradigm. The large model then verifies this sequence in a single forward pass, selecting a prefix to serve as part of the final output. This verification ensures that each sampled token adheres to the probability distribution parameterized by the large model. Since multiple tokens can be taken as the final output by only one forward pass of the large model, the acceleration can be achieved; **(B) Jacobi decoding**: Jacobi decoding deems the auto-regressive inference as a process of solving the fixed point of a nonlinear equation in a triangular system. This decoding algorithm iteratively performs multi-token decoding and can be executed without fine-tuning or auxiliary modules. The specific process of Jacobi decoding is as follows: First, given the previously pre-filled or decoded tokens, we randomly initialize a sequence of candidate tokens. Then, in each iteration, we execute one forward pass of the auto-regressive model for all the candidate tokens with a causal mask. the predicted probabilities then generate the tokens typically via greedy sampling, and these sampled tokens are taken as the inputs of the next iteration. This process can be formulated as: $x\_{i}^{(j+1)} = \arg \max\_x  p\_\theta(x|\mathbf{x}\_{1:i-1}^{(j)})$, where $i$ denotes the token index and $j$ denotes the iteration index. The Jacobi decoding process continues iterating until the convergence is reached, as determined by a deterministic criterion where these tokens remain unchanged between consecutive iterations.
>
>
>
> ## 2. **Investigations on the refinement**
>
> Thank you for your suggestion. Since we cannot include links in this rebuttal, we use a pair of tables to show the change of token categories at each sampling step and token position, so each table includes an example of the token trajectory. In SJD2 (shown by each column of Table A), over 25 sampling steps, the predicted tokens start identical (e.g., token value 167282). As the noise level decreases, the token categories become diverse, but some token categories begin to repeat, which means the trajectory becomes stabilize. In contrast, for SJD (the columns in Table B), the token categories change irregularly, appearing to oscillate and remain unstable. We will provide more detailed visualizations using figures in the camera-ready version.
>
> **Table A: The discrete token trajectory in SJD2 with Emu3 as baseline:**
>
> | Sampling Steps \   Token index | Token 1 | Token 2 | Token 3 | Token 4 | Token 5 |
> |---------------------------------|---------|---------|---------|---------|---------|
> | Step 1                          | 167282  | 167282  | 167282  | 167282  | 167282  |
> | Step 2                          | 167282  | 167282  | 167282  | 167282  | 167282  |
> | Step 3                          | 167282  | 167282  | 167282  | 167282  | 167282  |
> | Step 4                          | 167282  | 167282  | 167282  | 167282  | 167282  |
> | Step 5                          | 167728  | 167728  | 167728  | 167728  | 167728  |
> | Step 6                          | 167728  | 159459  | 159459  | 167728  | 159459  |
> | Step 7                          | 167282  | 167282  | 167282  | 167282  | 167282  |
> | Step 8                          | 167282  | 167282  | 163124  | 167282  | 152950  |
> | Step 9                          | 154915  | 153542  | 163124  | 156927  | 160877  |
> | Step 10                         | 155499  | 158046  | 158230  | 153542  | 152491  |
> | Step 11                         | 154578  | 157671  | 152153  | 153840  | 153399  |
> | Step 12                         | 154794  | 155092  | 152611  | 154282  | 153088  |
> | Step 13                         | 153692  | 153700  | 158037  | 156095  | 154951  |
> | Step 14                         | 152784  | 160040  | 156856  | 152551  | 155736  |
> | Step 15                         | **152388**  | 153700  | **155986**  | 152395  | 151927  |
> | Step 16                         | 158046  | 160040  | 165298  | 159796  | 151980  |
> | Step 17                         | **152388**  | *152357*  | 153737  | 155415  | *153182*  |
> | Step 18                         | 158046  | 160040  | **155986**  | **152006**  | *153182*  |
> | Step 19                         | **152388**  | *152357*  | **155986**  | **152006**  | **152268**  |
> | Step 20                         | 154500  | **155680**  | 153737  | 155493  | **152268**  |
> | Step 21                         | **152388**  | *152357*  | **155986**  | 154177  | **152268**  |
> | Step 22                         | 158046  | **155680**  | **155986**  | **152006**  | **152268**  |
> | Step 23                         | **152388**  | *152357*  | **155986**  | **152006**  | **152268**  |
> | Step 24                         | **152388**  | **155680**  | 153216  | **152006**  | 152519  |
> | Step 25                         | **152388**  | **155680**  | 153951  | 157603  | **152268**  |
>
>
> **Table B: The discrete token trajectory in SJD with Emu3 as baseline:**
>
> | Sampling Steps \   Token index | Token 1 | Token 2 | Token 3 | Token 4 | Token 5 |
> |---------------------------------|---------|---------|---------|---------|---------|
> | Step 1                          | 178678  | 154122  | 180405  | 156118  | 153692  |
> | Step 2                          | 165560  | 164685  | 162805  | 167164  | 155025  |
> | Step 3                          | 166380  | 154949  | 164287  | 166096  | 157596  |
> | Step 4                          | 156660  | 159530  | 152575  | 152503  | 166994  |
> | Step 5                          | 158407  | 151855  | 153356  | 157268  | 165140  |
> | Step 6                          | 160136  | 152594  | 160547  | 153105  | 152290  |
> | Step 7                          | 162159  | 156850  | 156404  | 159634  | 155860  |
> | Step 8                          | 165406  | 154488  | 158930  | 159965  | 153749  |
> | Step 9                          | 164959  | 151927  | 160406  | 165482  | 154677  |
> | Step 10                         | 154998  | 155867  | 166882  | 159530  | 166918  |
> | Step 11                         | 155948  | 159306  | 153403  | 156096  | 162431  |
> | Step 12                         | 160407  | 152980  | 155008  | 157592  | 152012  |
> | Step 13                         | 160393  | 156175  | 153776  | 156225  | 165458  |
> | Step 14                         | 155559  | 157817  | 164655  | 166620  | 159092  |
> | Step 15                         | 161548  | 153057  | 164198  | 155867  | 167001  |
> | Step 16                         | 152190  | 155639  | 164741  | 165587  | 151964  |
> | Step 17                         | 155733  | 163798  | 153648  | 163956  | 160777  |
> | Step 18                         | 152029  | 156397  | 154102  | 153091  | 155513  |
> | Step 19                         | 165985  | 152276  | 154563  | 159234  | 153776  |
> | Step 20                         | 161888  | 154110  | 167323  | 155040  | 154248  |
> | Step 21                         | 160647  | 152453  | 157631  | 153105  | 157318  |
> | Step 22                         | 161724  | 154949  | 159608  | **154419**  | 155366  |
> | Step 23                         | 155774  | 152065  | 164738  | 157863  | 157972  |
> | Step 24                         | 152548  | 153464  | 155459  | **154419**  | 156705  |
> | Step 25                         | 164009  | 158589  | 160031  | 153100  | 155766  |
>
> ## 3. **About the possibility of batch inference**
>
> In principle, our SJD2 can inherently support batch inference as a variant of speculative decoding [A]. However, our current implementation has not yet been optimized for batch processing due to engineering challenges such as synchronization overhead. We are actively exploring engineering optimizations, including dynamic batching and asynchronous verification, to enable efficient batch inference in future versions.
>
> [A] Qian H, Gonugondla S K, Ha S, et al. BASS: Batched attention-optimized speculative sampling[J]. arXiv preprint arXiv:2404.15778, 2024.
>
> ## 4. **About details of Equation 3**
> As shown in Appendix A, we set $\alpha_t=1-t$ and $\sigma_t=t$ for Equation 3.

---

> > ### Comment · Reviewer_3CPQ · 2025-08-05
> >
> > Thank you for your detailed and thoughtful rebuttal. I appreciate the additional empirical study you provided, which effectively addresses my initial concern about the behavior of denoising in discrete token space.

---

### Official Review · Reviewer_eMe4 · 2025-07-01

**Clarity:** 2
**Significance:** 3
**Originality:** 3
**Rating:** 4
**Confidence:** 4

**Summary:**

This paper proposed Speculative Jacobi-Denoising Decoding (SJD2) to accelerate autoregressive text-to-image model. By incorporating diffusion denoising with speculative decoding for AR model, the paper reaches 4x speed up on common AR image generation model.

**Questions:**

1. The index of embedding $ e $ in Eq.(2) is confused, please explain why the superscript skips from 0 to $ t_0 $, and what is $ i $ and $ i' $ in subscript.
2. Does the original AR model utilize normalized embedding as input?
3. Please explain the training objective and process, I only see a cross-entropy loss for AR model, how did this work for the diffusion? If there is a multi-step denoising process, would this involve a high-order gradient?
4. What is the finetuning effort of SJD2, like the dataset size and training time.
5. Please compare with more parallel decoding methods, like Jacobi Decoding, Lookahead Decoding, CLLMs, and ZipAR.
6. How effective is the Refinement with Denoising? Is the difference between SJD and SJD2 this refinement? If no, there is a lack of relevant ablation experiments.

**Ethical Concerns:**

["NO or VERY MINOR ethics concerns only"]

**Final Justification:**

My final rating is 4 Borderline accept. The authors proposed a decoding method to accelerate the AR image generation model, which shows benefits. My concerns are answered, the paper should be revised to make it clearer for audience.

**Limitations:**

See Weaknesses and Questions

**Quality:**

2

**Strengths And Weaknesses:**

**Strengths**
1. The background of SJD2 is well introduced in the first section.
2. SJD2 gets good acceleration with almost no loss on image quality.
3. The method works on both text-to-image and text-to-video models.

**Weaknesses**
1. Based as pervious work SJD, this work is only a minimal piece of contribution, besides, as stated in related work, the integration of diffusion has been studied in many visual generation works.
2. The method section is not well organized, needs more details, like the finetuning objective, the timestep injection and so on.
3. The difference between SJD2 and AR+diffusion methods are not introduced in related works.
4. The method comparison is not effective, only SJD are compared.

---

> ### Author Rebuttal · Authors · 2025-07-31
>
> # Response to Reviewer eMe4
>
> Thank you for your thoughtful and constructive feedback on our paper. We appreciate your recognition of our introduction and the significant acceleration.
>
> ## **1. About our contribution**
>
> We feel it necessary to clarify the contribution of our work: our SJD2 introduces a new advancement by integrating denoising process into autoregressive (AR) models to **stabilize the Jacobi iteration in SJD**. We achieve this through **fine-tuning** AR models to be **compatible with noisy embeddings** (thus the denoising can be integrated in Jacobi iteration, shown in Eq. 3, lines 191-202), while distinctively **preserving core AR mechanisms** (discrete tokenization, next-token prediction with cross-entropy loss, and Jacobi iteration) via the proposed techniques like **noise perturbation on normalized embeddings**. Unlike existing AR+diffusion models that rely on diffusion losses and auxiliary decoders for image generation, SJD2 **maintains the standard AR components for image generation** (i.e. **predicting the probability of discrete tokens**, and **training with cross-entropy loss**) while delivering significant acceleration gains (shown in Table 1).
>
> ## **2. About details of our finetuning objective and timestep injection**
>
> Thank you for your suggestion. We will improve the clarity of the method section in the camera-ready version and we detail our finetuning objective and timestep injection here: **(A) timestep injection**: Since we avoid introducing additional adapters, we utilize a flexible operator, i.e., the attention mechanism, for our token-wise timestep injection. Specifically, we take the sinusoidal encodings of timesteps as a sequence of special token embeddings and append them to the sequence of input token embeddings. Then, the sequence, which comprises clean input token embeddings, noisy input token embeddings, and timestep encodings, is fed into the transformer blocks. Within the attention modules of these blocks, we use the attention mask to force each noisy token embedding to attend to the corresponding timestep encoding which indicates its noise level. To ensure the distribution of the Fourier encodings of timesteps aligns with that of the token embeddings, we apply a normalization-then-de-normalization process to these encodings (normalization with statistics of the sinusoidal encodings and denormalization with the statistics of the token embeddings), similar to Eq 4. **(B) finetuning objective**: SJD2 fine-tunes a pre-trained autoregressive text-to-image model to predict the next clean token from noisy token embeddings. During fine-tuning, input token embeddings are perturbed with Gaussian noise and then fed into transformer blocks followed by a prediction head which outputs logits representing the categorical probability distribution of the next clean token. The cross-entropy loss is computed between these logits and the ground truth token categories, optimizing the model to denoise inputs while maintaining autoregressive prediction. This objective enables SJD2 to handle noisy inputs during inference.
>
>
> ## **3. Difference between SJD2 and AR+diffusion methods**
>
> In our related work section (Integration of autoregression and continuous diffusion), we have discussed models like Diffusion-forcing and Transfusion. However, with the recent publication of numerous new AR+Diffusion works, we now provide an updated version of the related work part: Recent diffusion forcing models (e.g. Self-forcing [A], MAGI-1 [B], and Causvid [C]) introduce the autoregressive sampling into the temporal dimension of the continuous video diffusion models. They employ causal masks to support history-conditioned video generation. The multimodal large language models (MLLMs) (e.g. BLIP-3o [D], Emu2 [E], and Seed-X [F]) generate images via pre-trained diffusion models (e.g., SDXL) which take the output features of autoregressive models as conditions. Moreover, the unified models like Bagel [G] and Janus-flow [H] directly integrate diffusion process into the autoregressive backbones. These models train the autoregressive backbones together with lightweight decoders through diffusion loss (e.g., noise prediction or velocity prediction) for image generation, and the backbone can perform both autoregressive decoding and continous diffusion sampling. In contrast, SJD2 uniquely integrates denoising into pre-trained autoregressive models without modifying their core: **preserving discrete tokenizers, next-token prediction with cross-entropy loss, and the established inference mechanisms**. Our innovation lies in **fine-tuning with noise-perturbed normalized embeddings**, enabling ODE-based denoising in autoregressive models. We do **not** relay on diffusion loss and additional decoders.
>
> [A] Huang X, Li Z, He G, et al. Self Forcing: Bridging the Train-Test Gap in Autoregressive Video Diffusion[J]. arXiv 2025.
>
> [B] Teng H, Jia H, Sun L, et al. MAGI-1: Autoregressive Video Generation at Scale[J]. arXiv 2025.
>
> [C] Yin T, Zhang Q, Zhang R, et al. From slow bidirectional to fast autoregressive video diffusion models[C]. CVPR 2025.
>
> [D] Chen J, Xu Z, Pan X, et al. Blip3-o: A family of fully open unified multimodal models-architecture, training and dataset[J]. arXiv 2025.
>
> [E] Sun Q, Cui Y, Zhang X, et al. Generative multimodal models are in-context learners[C]. CVPR 2024.
>
> [F] Ge Y, Zhao S, Zhu J, et al. Seed-x: Multimodal models with unified multi-granularity comprehension and generation[J]. arXiv 2024.
>
> [G] Deng C, Zhu D, Li K, et al. Emerging properties in unified multimodal pretraining[J]. arXiv 2025.
>
> [H] Ma Y, Liu X, Chen X, et al. Janusflow: Harmonizing autoregression and rectified flow for unified multimodal understanding and generation[C]. CVPR 2025.
>
>
> ## **4. Comparison to modern parallel decoding methods**
>
> We compare our method with the recent state-of-the-art and classic speculative/parallel decoding methods, including Lantern (ICLR 2025) [A], ZipAR (ICML 2025) [B], Eagle [C] and Jacobi Decoding [D], on COCO2017 validation set with Lumina-mGPT as baseline. As shown in the table below, our approach achieves superior acceleration while maintaining comparable visual quality.
>
> | Configuration | Acceleration Latency $(\uparrow)$ | Acceleration Step $(\uparrow)$ | CLIP-Score $(\uparrow)$ |
> |---------------|------------------|------------------|---------|
> |  AR | $1.00 \times$ | $1.00 \times$ | $31.3$ |
> |  Jacobi Decoding | $1.02 \times$ | $1.04 \times$ |$31.4$ |
> |  SJD | $2.05 \times$ | $2.23 \times$ |  $31.3$ |
> | EAGLE | $2.10 \times$ | $2.94 \times$ | $33.3$ |
> |  LANTERN | $2.56 \times$ | $3.63 \times$ | $32.7$ |
> |  ZipAR | $1.82 \times$ | $4.00 \times$ |$31.2$ |
> |  Ours | $2.63 \times$ | $4.02 \times$ |  $31.8$   |
>
> [A] Jang D, Park S, Yang J Y, et al. Lantern: Accelerating visual autoregressive models with relaxed speculative decoding[C]. ICLR 2025.
>
> [B] He Y, Chen F, He Y, et al. ZipAR: Parallel Autoregressive Image Generation through Spatial Locality[C]. Forty-second International Conference on Machine Learning (ICML 2025).
>
> [C] Li Y, Wei F, Zhang C, et al. EAGLE: Speculative Sampling Requires Rethinking Feature Uncertainty[C]. Forty-first International Conference on Machine Learning (ICML 2024).
>
> [D] Song Y, Meng C, Liao R, et al. Accelerating feedforward computation via parallel nonlinear equation solving[C]. PMLR 2021.
>
>
> ## **5. About the superscript and subscript in equation 2**
>
> The superscript in Equation (2) indicates the noise level applied to the embedding $e$, with the skip from $0$ to $t_0$ reflecting a deliberate design in the application of noise perturbation. As demonstrated in Figure 3, some regions of the embeddings remain unperturbed by noise (denoted by superscript 0), while their adjacent regions are perturbed to a minimal noise level (denoted by superscript $t_0$). The subscript $i$ identifies the position of a specific token embedding, and $i'$ is an integer in the range $[1, i]$, used to specify a certain position where the noise level shifts from $0$ to $t_0$.
>
> ## **6. About inputs in original autoregressive models**
>
> The pre-trained autoregressive models used as baselines are incompatible with normalized embedding inputs. Consequently, after performing denoising on normalized embedding inputs, we employ a **denormalization** step (Equation 4) to restore these noisy normalized embeddings to the original embedding space.
>
>
> ## **7. About the training process**
>
> Unlike standard diffusion models that rely on diffusion loss (i.e. $\mathbb{E}\_{t,\epsilon,x_0} [ ||v\_\theta(x\_t,t)-(\epsilon-x\_0)|| ]$), the training objective of SJD2 is based **solely on cross-entropy loss**. This is because SJD2 is **not** a standard diffusion model: it only **leverages the denoising concept from diffusion models** while **not** using the diffusion loss. Specifically, during training, we perturb token embeddings with noise and the cross-entropy loss is applied on the noisy embeddings. During inference, the ODE denoising process is fused with the Jacobi decoding which is also a multi-step decoding algorithm (line 191-202 with equation 3). There is no high-order gradient computations in our fine-tuning.
>
>
> ## **8. About the fine-tuning efforts**
>
> To build the training dataset for fine-tuning, we collect about 80,000 synthesized images from huggingface and recaption them with Qwen-VL. We perform fine-tuning on 8 NVIDIA GPUs (80GB memory each) with a global batch size of 64, while leveraging DeepSpeed ZeRO-3 or FSDP with gradient checkpointing to save GPU memory at the cost of increased training time. All model parameters are used for fine-tuning. The fine-tuning requires 6 epochs, which costs about $14 \times 8$ A100 hours for Lumina-mGPT and $26 \times 8$ H100 hours for Emu3.
>
> ## **9. About the refinement with denoising**
>
> This refinement with denoising is the key distinction between SJD2 and SJD. As shown in Table 1, SJD2 achieves greater acceleration than SJD.

---

> ### Comment · Reviewer_eMe4 · 2025-08-05
> **Thanks for the rebuttal**
>
> Thanks for the rebuttal, my questions are partially answered. The author should make the proposed method (AR & Diffusion) clearer in paper, both training & inference procedure, text description & figures (like Fig.3).

---

> ### Comment · Reviewer_eMe4 · 2025-08-05
> **Question about denoising steps.**
>
> To my view, the proposed method is to predict next token until the noisy embedding is acceptable. If so, I'd like to know how many denoising step would it take during the generation of each token?

---

> > ### Author Response · Authors · 2025-08-05
> > **Reply to Reviewer eMe4**
> >
> > We appreciate your feedback and are pleased that the previous concerns have been addressed. We will incorporate the improvements to enhance the clarity of our methodology in the final version of the paper. Here, we will address the new concerns:
> >
> > ## **About the illustration of training and inference procedure**
> >
> > Thank you for your suggestion. We will improve clarity of writing in our method section. Here, we **explain the specific training and inference procedures** of SJD2 using Figure 2 and Figure 3, with the following text describing the figures:
> >
> > **Figure 2**: Overview of our **decoding** process. *First*, given a sequence of noisy normalized token embeddings (illustrated as blue-bordered patches in the first row) and prefilling/already-accepted tokens (depicted as green circles in the first row), the noise levels of the token embeddings are set to increase non-strictly monotonically. *Next*, these token embeddings undergo one iteration of Jacobi-denoising, where they are fed into the neural network along with timestep encodings for a single parallel forward pass using a causal attention mask. The network predicts conditional probability and performs token sampling for the next clean token at each position. Sampled tokens from noisy embedding inputs are denoted as the one-position-offset $\hat{x}^{0}$-predictions, marked by the down-right blue solid arrows, while those from prefilling or accepted token inputs are denoted as autoregressive (AR) predictions (i.e. the standard next-token prediction), marked by green solid arrows. *Subsequently*, a prefix of sampled tokens is accepted based on the probabilistic criterion outlined in Equation 1 (e.g., in the second row, the first two sampled tokens are accepted and marked with green borders). Then, a denoising step (Equation 3) is performed for the remaining tokens, involving a linear combination of the token embedding from the previous iteration at the same spatial position (indicated by vertical blue dashed lines) and the embedding of the predicted clean tokens from the one-position offset (the down-right blue solid arrows). This iterative process repeats until all required tokens are accepted, serving as the final outputs.
> >
> >
> > **Figure 3**: This figure illustrates our **training** strategy and **model** process. Starting with normalized input embeddings rather than raw input indices (i.e., token categories), the process is depicted as follows: **(a) During training**, the normalized embeddings (initially clean) are first transformed into noisy embeddings, as shown in the right-side image. In this image, noise levels increase non-monotonically across patches (i.e., token positions) in a raster scan order (left to right, top to bottom). When reaching a randomly determined position, the noise level stops increasing, and the noise level of the next position resets to zero, forming segments with non-monotonically increasing noise levels in the token sequence. Next, these noisy embeddings are appended with timestep encodings, which indicate the noise level at each position. Together, the embeddings and encodings are fed into transformer blocks and a prediction head to produce logits for each position. The cross-entropy loss is then applied to each position, using the clean token indices as labels, with one-position offset (shown by the pink dotted frame) for next-clean-token prediction. **(b) During inference**, the input normalized embeddings (already noisy) are not further perturbed. These embeddings are also appended with timestep encodings and processed through the transformer blocks and prediction head to generate logits. As described in Equation 2, these logits are used for token sampling, and these sampled clean tokens are transformed into normalized token embeddings, consistent with the input normalized embeddings shown in this figure.
> >
> > ## **About the denoising process in the procedure of token prediction**
> > Yes. Given a sequence of token embeddings, SJD2 performs the next-token prediction for each embedding until its corresponding token category is acceptable. The number of denoising steps in SJD2 is set to 25.

---

### Official Review · Reviewer_Q75X · 2025-07-01

**Clarity:** 3
**Significance:** 2
**Originality:** 3
**Rating:** 4
**Confidence:** 4

**Summary:**

This paper introduces Speculative Jacobi-Denoising Decoding (SJD2), a method for enabling parallel token generation in autoregressive models. The authors propose a next-clean-token-prediction task that allows a model to predict multiple future tokens with only light fine-tuning. Experiments on MSCOCO and GenEval show that this approach can achieve a speedup of around 4x without a significant drop in performance.

**Questions:**

- Why was there no comparison with the Emu3 model on the GenEval benchmark, even though it was used in other experiments?

- Are there any comparisons of computational cost in terms of FLOPs? This would provide a more detailed efficiency analysis beyond latency.

- How does the proposed SJD2 approach compare to other parallel decoding methods from recent literature? A broader comparison is needed to position this work.

- Is SJD2 compatible with autoregressive models that already use a form of parallel decoding, such as MAR models?

- Could you provide an analysis or comparison of the training cost required for the proposed SJD2 approach?

**Ethical Concerns:**

["NO or VERY MINOR ethics concerns only"]

**Final Justification:**

The author's rebuttal resolves most of my concerns and I'm willing to increase the score. Authors are encouraged to update the manuscript to reflect the comparison with more baselines.

**Limitations:**

Yes

**Quality:**

3

**Strengths And Weaknesses:**

Strengths

- The paper is well-written, and the ideas and methods are described clearly. The main figure illustrates the core idea very well.

- The evaluation on MSCOCO shows that the method can speed up models like Lumina-mGPT and Emu3 by about 4x without harming performance too much.

- The qualitative results look good and effectively support the main experimental claims.

Weaknesses

- Limited Scope of Model Compatibility: It is unclear if the proposed SJD2 method works with a wider range of autoregressive models. For example, compatibility with other advanced autoregressive approaches like VAR or Janus Pro is not discussed.

- Insufficient Baseline Comparisons: The main experiments lack comparisons with other modern speculative or parallel decoding methods. The only baseline compared against is the original SJD, which makes it hard to judge SJD2's performance relative to the state of the art.

- Missing Comparison with Diffusion Models: The paper does not include a comparison of the efficiency/performance trade-off against state-of-the-art diffusion models.

- Weak Performance on GenEval: The overall performance on the GenEval benchmark does not appear to be competitive with many existing methods.

---

> ### Author Rebuttal · Authors · 2025-07-31
>
> # Response to Reviewer Q75X
>
>
> Thank you for your thoughtful and constructive feedback on our paper. We particularly appreciate your recognition of the clear methodological presentation and our performance of acceleration. Our primary contribution lies in enabling efficient parallelization of existing autoregressive architectures through lightweight fine-tuning, rather than pursuing state-of-the-art performance.
>
>
> ## **1. Results on more autoregressive models**
>
> Thank you for your suggestion. As our SJD2 is developed for the paradigm of next-token prediction, we choose Janus-pro-1B for experiments. The paradigm of VAR is next-scale prediction which could require different ways of acceleration and thus is beyond the scope of our paper. Specifically, we employ our pairwise text-image data to tune Janus-pro-1B to be compatiable with noisy embedding inputs and use our SJD2 for decoding. The results are in the following tables. From the results, we observe that SJD2 still can accelerate Janus-pro without sacrificing on generated image quality, as evidenced by the following Geneval metrics.
>
> | Method | Latency | Steps |
> |---------|---------------|------------------|
> | Janus-Pro-1B | 9.1s | 576 |
> | Janus-Pro-1B + SJD2 | 2.5s | 144  |
>
> | Method | Colors | Position | Counting | Two | Color Attri| Single | Overall |
> |---------|---------------|------------------|------------------|---------|----------------|----------------|----------------|
> | Janus-Pro-1B (AR) |  0.82 | 0.39   |  0.39 | 0.55  |  0.42 | 0.94 | 0.59  |
> | Janus-Pro-1B (SJD2) | 0.83 | 0.45   | 0.37  | 0.59  | 0.38  |0.96 | 0.60 |
>
> ## **2. Comparison to modern parallel decoding methods**
> We compare our method with the recent state-of-the-art and classic speculative/parallel decoding methods, including Lantern (ICLR 2025) [A], ZipAR (ICML 2025) [B], Eagle [C] and Jacobi Decoding [D], on COCO2017 validation set with Lumina-mGPT as baseline. As shown in the table below, our approach achieves superior acceleration while maintaining comparable visual quality.
>
> | Configuration | Acceleration Latency $(\uparrow)$ | Acceleration Step $(\uparrow)$ | CLIP-Score $(\uparrow)$ |
> |---------------|------------------|------------------|---------|
> |  AR | $1.00 \times$ | $1.00 \times$ | $31.3$ |
> |  Jacobi Decoding | $1.02 \times$ | $1.04 \times$ |$31.4$ |
> |  SJD | $2.05 \times$ | $2.23 \times$ |  $31.3$ |
> | EAGLE | $2.10 \times$ | $2.94 \times$ | $33.3$ |
> |  LANTERN | $2.56 \times$ | $3.63 \times$ | $32.7$ |
> |  ZipAR | $1.82 \times$ | $4.00 \times$ |$31.2$ |
> |  Our SJD2 | $2.63 \times$ | $4.02 \times$ |  $31.8$   |
>
> [A] Jang D, Park S, Yang J Y, et al. Lantern: Accelerating visual autoregressive models with relaxed speculative decoding[C]. ICLR 2025.
> [B] He Y, Chen F, He Y, et al. ZipAR: Parallel Autoregressive Image Generation through Spatial Locality[C]. Forty-second International Conference on Machine Learning (ICML 2025).
> [C] Li Y, Wei F, Zhang C, et al. EAGLE: Speculative Sampling Requires Rethinking Feature Uncertainty[C]. Forty-first International Conference on Machine Learning (ICML 2024).
> [D] Song Y, Meng C, Liao R, et al. Accelerating feedforward computation via parallel nonlinear equation solving[C]. PMLR 2021.
>
>
> ## **3. About the lack of comparison to diffusion models**
>
> Actually, our work focuses on improving the inference efficiency of standard autoregressive (AR) models for image generation. Thus, we choose these baselines: Lumina-mGPT, Emu3, and Janus-Pro that is supplemented in this rebuttal. The AR models have advantages in the unified modeling of multimodal data (predicting discrete tokens for both the linguistic and visual tasks, different from the diffusion models) and thus it is worth studying [A, B, C]. While many AR models currently underperform state-of-the-art diffusion models in image quality and face acceleration challenges, our SJD2 narrows the speed gap. In the following table, we evaluate inference latency for several commonly-used diffusion models (smaller than 3B) and Janus-Pro at the same resolution ($384 \times 384$). We set the number of sampling steps for diffusion models as 50. Results demonstrate that our SJD2 reduces latency of Janus-Pro-1B from 9.1s to 2.5s, narrowing the gap between Janus-pro and the advanced diffusion models like SD3. Moreover, this result means Janus-Pro-1B with SJD2 already outperforms SDXL in speed (2.5s vs. 4.3s):
>
> | Method | Latency | Steps |
> |---------|---------------|------------------|
> | Janus-Pro-1B | 9.1s | 576 |
> | SD1.5 | 1.7s |  50  |
> | SDXL | *4.3s* | 50  |
> | SD3-Medium | *1.7s* | 50  |
> | Janus-Pro-1B + SJD2 | *2.5s* | 144  |
>
> [A] Chen X, Wu Z, Liu X, et al. Janus-pro: Unified multimodal understanding and generation with data and model scaling[J]. arXiv preprint arXiv:2501.17811, 2025.
>
> [B] Wang X, Zhang X, Luo Z, et al. Emu3: Next-token prediction is all you need[J]. arXiv preprint arXiv:2409.18869, 2024.
>
> [C] Team C. Chameleon: Mixed-modal early-fusion foundation models[J]. arXiv preprint arXiv:2405.09818, 2024.
>
> ## **4. Regarding Performance on GenEval**
> This work focuses on **accelerating existing pre-trained autoregressive models while preserving visual quality**, not on achieving state-of-the-art image quality. As shown in Table 2, our method successfully maintains visual fidelity on GenEval. While current autoregressive baselines like Lumina-mGPT generally underperform compared to advanced diffusion-based generation models, our proposed method has the potential to be applied to stronger autoregressive base models in the future.
>
>
> ## **5. Performance of Emu3 on Geneval benchmark**
>
> In the following table, we compare our SJD2 to autoregressive decoding (AR) and SJD on the GenEval benchmark with Emu3 as the baseline. Our method achieves an overall score of 0.49, which is close to the 0.52 of AR and the 0.48 of SJD, demonstrating preserved visual quality.
>
> | Method | Colors | Position | Counting | Two | Color Attri| Single | Overall |
> |---------|---------------|------------------|------------------|---------|----------------|----------------|----------------|
> | AR | 0.78 | 0.15   | 0.33  | 0.69  | 0.16  |0.98 | 0.52 |
> | SJD | 0.79 | 0.12   | 0.28  | 0.61  | 0.13  |0.97 | 0.48 |
> | SJD2 | 0.73 | 0.14  | 0.28  | 0.61 | 0.24  |0.96 | 0.49 |
>
>
> ## **6. Flops in inference**
>
> We present the average Flops per output token in the table below. The results reveal that autoregressive decoding requires fewer Flops than SJD2. *Although more Flops are used for decoding, the practical latency becomes lower*. Actually, this Flops overload stems from the paradigm of all the speculative decoding methods. Their drafting-and-verification mechanism, which inherently introduces computational overhead: the number of accepted tokens per sampling step is substantially lower than the number of input draft tokens. Previous studies have also confirmed this observation. For example, as demonstrated in Tables 4 and 5 of [A], Medusa (a speculative decoding method) consumes 403 GFlops per token, whereas vanilla autoregressive decoding requires only 19 GFlops.
>
> | Method | GFlops |  Latency|
> |---------|---------------|-------------|
> | Lumina-mGPT (AR) | 18.72 | 88.55s |
> | Lumina-mGPT (SJD2) | 219.60 | 33.64s |
> | Emu3 (AR) | 18.15 |  375.29s |
> | Emu3 (SJD2) |  465.92 | 147.65s |
>
> [A] Lin C H, Tuli S, Smith J, et al. SLiM: Speculative decoding with hypothesis reduction[C]//Findings of the Association for Computational Linguistics: NAACL 2024. 2024: 1005-1017.
>
>
>
> ## **7. About compatibility of parallel-decoding models**
> Thank you for your question. Actually, speculative decoding was originally designed to accelerate standard autoregressive models that decode tokens sequentially (one token per forward pass). Its lossless acceleration relies on optimizing GPU parallelism utilization. However, the models employing parallel decoding techniques (e.g., MaskGit or the next-set-prediction of MAR) inherently utilizes GPU parallelism well. Consequently, speculative decoding methods like SJD2 would require redesign and new theoretical foundations to assist these frameworks, which can be explored for future works. Regarding the diffusion head of MAR which operates on continuous latents, the speculative decoding should be redesigned to compatiable with Gaussian inputs, and some concurrent efforsts have been conducted and still in exploration [A].
>
> [A] Wang Z, Zhang R, Ding K, et al. Continuous speculative decoding for autoregressive image generation[J]. arXiv preprint arXiv:2411.11925, 2024.
>
> ## **8. Comparison of the training cost of SJD2**
> Regarding the fine-tuning cost, the baseline, Lumina-mGPT, requires 10 million image-text pairs for pre-training. In contrast, our fine-tuning uses only 80,000 images over 6 epochs, totaling approximately 0.5 million image samples used during training. This indicates that at most 5% of the original computational resources are used for fine-tuning our method. We only use $14\times8$ A100 hours for fine-tuning, highlighting the efficiency.

---

> > ### Comment · Reviewer_Q75X · 2025-08-06
> >
> > Thanks the author for the detailed feedback. SJD2 seems to have good performance on token-by-token autoregressive vision models.
> > However, I'm still having concerns about it not generalizable to more autoregressive approaches, Also the numbers for Emu3 on GenEval seems to be lower than the ones reported in the original paper.

---

> > > ### Author Response · Authors · 2025-08-07
> > > **Reply to Reviewer Q75X**
> > >
> > > Thank you for your feedback. We will address the new concerns here:
> > >
> > > ## **About the number of autoregressive base models**
> > >
> > >
> > > In this paper and our prior rebuttal, we have conducted experiments on *three* base models: Lumina-mGPT, Emu3, and Janus-Pro. Following the suggestions in the prior reviews, we included Janus-Pro to demonstrate the effectiveness of SJD2 on the advanced autoregressive models. Notably, existing papers on accelerating autoregressive text-to-image generation typically use 2–3 base models [A, B] to validate generalizability, aligning with the number of autoregressive base models for SJD2.
> > >
> > > [A] He Y, Chen F, He Y, et al. ZipAR: Parallel Autoregressive Image Generation through Spatial Locality[C]. Forty-second International Conference on Machine Learning.
> > >
> > > [B] Jang D, Park S, Yang J Y, et al. LANTERN: Accelerating Visual Autoregressive Models with Relaxed Speculative Decoding[C]. The Thirteenth International Conference on Learning Representations.
> > >
> > > ## **About the GenEval performance on Emu3**
> > >
> > > First, the baseline performance of Emu3 with original autoregressive decoding (AR), as shown in the first row of the table of GenEval-Emu3 (reproduced here from our prior rebuttal), aligns with the original Emu3 paper (Page 21, Table 7, second-to-last line, Emu3-DPO, overall score: 0.52) [A]. Second, our SJD2 method achieves a GenEval overall score of 0.49, closely approaching the performance of Emu3 with AR decoding (0.52). Furthermore, the speculative decoding method SJD [B] scores 0.48, demonstrating that our approach is competitive with existing acceleration methods.
> > >
> > > | Method | Colors | Position | Counting | Two | Color Attri| Single | Overall |
> > > |---------|---------------|------------------|------------------|---------|----------------|----------------|----------------|
> > > | AR | 0.78 | 0.15   | 0.33  | 0.69  | 0.16  |0.98 | 0.52 |
> > > | SJD | 0.79 | 0.12   | 0.28  | 0.61  | 0.13  |0.97 | 0.48 |
> > > | SJD2 | 0.73 | 0.14  | 0.28  | 0.61 | 0.24  |0.96 | 0.49 |
> > >
> > > [A] Wang X, Zhang X, Luo Z, et al. Emu3: Next-token prediction is all you need[J]. arXiv preprint arXiv:2409.18869, 2024.
> > >
> > > [B] Teng Y, Shi H, Liu X, et al. Accelerating Auto-regressive Text-to-Image Generation with Training-free Speculative Jacobi Decoding[C]. The Thirteenth International Conference on Learning Representations.

---

### Official Review · Reviewer_ucHV · 2025-07-03

**Clarity:** 4
**Significance:** 3
**Originality:** 4
**Rating:** 6
**Confidence:** 4

**Summary:**

Speculative Jacobi-Denoising Decoding is the next step in proposing novel improvements to Speculative Jacobi Decoding for parallel token generation in autoregressive models. These changes result in faster convergence, reducing computational overhead, as well as equivalent or higher quality on several metrics demonstrated on image generation tasks.

**Questions:**

Do the authors have any user preference A/B tests to complement the GenEval result? It would be great to correlate these quantitative metrics to human A/B.

In eq(1), if the 1 in the lefthand side is decreased, what happens to the resulting quality? Do the authors have any study, reference, or ablation on this - I think it might also improve the convergence speed of Speculative Jacobi Decoding, but maybe at the expense of quality?

Do the authors have any high-order statistics, like skewness, kurtosis, and so on about the distribution of embeddings before and after normalization? It is slightly surprising to me that simple mean standard deviation normalization is enough to Gaussianize sufficiently to prevent the failure modes discussed in the paper. It would be good to have some information, if people want to apply this method but have different embedding distributions up front.

Is there any potential of a "training free" version of this method, if not what are the key roadblocks (technical, computational, theoretical) to such a version?

Will the authors release code to help reproduction in open source tools?

**Ethical Concerns:**

["NO or VERY MINOR ethics concerns only"]

**Final Justification:**

I like this paper and think it has large applicability with broad interest for a variety of autoregressive modeling approaches. Given the focused scope of the work and the clarity of discourse, I think it should be a "strong accept" for any conference. I really like the clear explanation and practicality of the method.

**Limitations:**

Yes

**Quality:**

4

**Strengths And Weaknesses:**

Strengths:
Detailed ablations and metric studies, as well as a clear description of the approach. I think this is a good, informative paper which is useful for people to get introduced to Jacobi decoding methods, as well as see the line of research leading up to this new method. The experimental section is focused on images, but clear in the results and goals. Descriptions of the inference time procedure are high quality, and presented with great figures and examples.

Weaknesses:
Given the widespread applicability of Jacobi decoding and descendants, it would be really interesting to see this approach applied to other generative applications, like text or audio token generation. It may be out of scope for this review cycle, but really worth considering for broad appeal of the method.

It is not clear to me if the full model is trained during the finetuning procedure, or only a small part. There are some other complaints below, but a bit more indepth treatment of the tuning stage, and what parameters are involved would be useful especially for followup work.

There is a strong conceptual convergence between the approach detailed here, and the paper "Rolling Diffusion Models", which adds a rolling time-window approach into continuous diffusion. It it is probably worth referencing or briefly discussing in the background material.

There was some mention of compute costs, but it would be useful to detail how long the finetuning takes in terms of approximate wall-clock time, as well as further details on the finetune hyperparameters in general.

There is a statement "For fine-tuning, we divide the input sequence into randomly sized segments and add noise to each segment.", it would be useful to have details about the procedure to select the randomly sized adjacent segments, and how the noise level is chosen and applied over training. There is "We follow the advanced flow matching setting with αt = 1 − t and σt = t for our denoising process", but more details would be better.

Similarly, there is a statement "we take the normalized Fourier encodings of timesteps" - is this different than a standard positional encoding? If so, it would be good to add more detail, or cite a relevant paper defining this. Otherwise, change the writing slightly to tie in to the more standard naming with "Fourier positional encodings" or "Transformer sinusoidal position encodings", or something like this.

---

> ### Author Rebuttal · Authors · 2025-07-31
>
> # Response to Reviewer ucHV
>
> Thank you for your thoughtful and constructive feedback on our paper. We particularly appreciate your recognition of our detailed ablation studies, clear methodological explanations, and high-quality experimental demonstrations.
>
> ## **1. About potential application on other domains like text/audio**
>
> Thank you for your thoughtful suggestion about extending Jacobi decoding to text and audio generation. While such expansion falls beyond the scope of current review cycle, we plan future research to pursue this direction, including curating diverse text/audio datasets and exploring integrations with autoregressive LLMs like Llama and Qwen by using SJD2.
>
>
> ## **2. About fine-tuning details and the training/inference cost**
> All model parameters are used for fine-tuning. To build the dataset, we collect about 80,000 synthesized images from huggingface and recaption them with Qwen-VL. We perform fine-tuning on 8 NVIDIA GPUs (80GB memory each) with a global batch size of 64, a learning rate of 2e-5, and the AdamW optimizer ($\beta_1$=0.9, $\beta_2$=0.95), while leveraging DeepSpeed ZeRO-3 or FSDP with gradient checkpointing to save GPU memory at the cost of increased training time. The fine-tuning requires 6 epochs, which costs about $14 \times 8$ A100 hours for Lumina-mGPT and $26 \times 8$ H100 hours for Emu3. The inference cost is in the following table:
>
> | Method | GPU memory | Latency|
> |---------|---------------|---------------|
> | Lumina-mGPT (AR) | 17G | 88.55s |
> | Lumina-mGPT (SJD2) | 20G  | 33.64s |
> | Emu3 (AR) | 20G | 375.29s |
> | Emu3 (SJD2) | 23G  | 147.65s |
>
> ## **3. About discussion on Rolling Diffusion Models**
>
>
> Thank you for your suggestion. SJD2 accelerates autoregressive text-to-image generation by integrating denoising with Jacobi iterations, using a fixed sliding window for parallel token prediction in discrete token space. It fine-tunes pre-trained models to predict conditional probability given the input of noisy token embeddings, and uses the cross-entropy loss for supervision. During inference, the token embedding is first denoised and then seamlessly enters the standard Speculative Jacobi iterations draft and verify its corresponding discrete token and probability. In contrast, Rolling Diffusion Model applies a continuous diffusion process with a rolling time-window for time-series data, trained with diffusion loss to predict noise in continuous space. While both use window-based denoising for efficiency, SJD2 focuses on the discrete tokens in autoregressive models, whereas Rolling Diffusion Model focuses on adapting continuous diffusion for temporal sequences. We will add the discussion in the camera-ready version.
>
>
> ## **4. About segment size and noise level selection**
> The process of sequence segment dividing and noise level selection is as follows: First, we sample a monotonically increasing timestep sequence $( 0, t\_{0}, \cdots, t\_{K'} , 1 )$ from the range of $[0,1]$, where $K'$ is randomly determined in range $[10, 50]$ (the typical number of steps in diffusion sampling). The elements $t\_i$ in this timestep sequence can adhere to the Karras timestep schedule [A]. Then, for an input token sequence of length $N$ ($N > K'$), we uniformly partition it into $n$ segments ($n$ is randomly chosen from range $[1, \frac{N}{K'}]$), and fill each segment uniformly with the timestep sequence $( 0, t\_{0}, \cdots, t\_{K'} , 1 )$ across all positions. These timesteps are used to calculate the parameters $\alpha_t=1-t$ and $\sigma_t=t$ for noise perturbation on token embeddings.
>
> [A] Karras T, Aittala M, Aila T, et al. Elucidating the design space of diffusion-based generative models[J]. Advances in neural information processing systems, 2022, 35: 26565-26577.
>
> ## **5. About the writting of positional encoding**
>
> Thank you for your suggesion. Our position encoding is the same as the sinusoidal position encodings used in standard Transformers. We will add this explanation in the camera-ready version.
>
> ## **6. About the user preference A/B tests**
>
> We conduct user preference on the images generated by autoregressive decoding and SJD2. We use Lumina-mGPT as baseline and select 16 pairs of images with 10 users for study. The results are in the following table. We observe that the preference of the two decoding methods is close and the images generated by autoregressive decoding is slightly more preferred.
>
> | Method  | Preference |
> |---------|--------------|
> | AR | 54.7\% |
> | SJD2 | 45.3\%   |
>
> ## **7. The influence of decreasing the 1 in the lefthand side in Equation 1**
>
> Thank you for your question. First, for simplicity, let's consider replacing the constant "1" on the left-hand side of Equation 1 with a variable $k$. Actually, decreasing this variable $k$ would reduce the acceptance rate, as a lower $k$ makes the acceptance condition $r <  \min \left( k,  \frac{\mathcal{P}\_\theta(x\_i^{(j)} | x\_{1}^{(j)}, \cdots, x\_{i-1}^{(j)})}{\mathcal{P}\_\theta(x\_i^{(j)} |  x\_{1}^{(j')}, \cdots, x\_{i-1}^{(j')}  )}  \right) , ~~ r \sim   \mathcal{U}[0,1]$ harder to satisfy. We also conduct corresponding experiments and the results aligns with our hypothesis. Thus, $k=1.0$ not only conforms to the theory of speculative sampling, but also guarantees a relatively large acceptance rate.
>
> | Method  | Steps |
> |---------|--------------|
> | Emu3 + SJD ($k=1.0$) | 3537 |
> | Emu3 + SJD ($k=0.5$) | 4969   |
>
> ## **8. About relationship between normalization and the failure mode**
>
> We do not utilize high-order statistics like skewness or kurtosis for normalization and denormalization. Actually, the failure mode depicted in Figure 5 arises when the noise-perturbation-based fine-tuning (and the denoising sampling) applied *directly* to token embeddings *without normalization and denormalization*. As we discussed in lines 213-220, normalization and denormalization primarily serve to *scale* input values appropriately for noise perturbation and transformer blocks, respectively. The noise perturbation formula $x\_t = \alpha\_t \cdot x\_0 + \sigma\_t \cdot \epsilon$, where $\epsilon \sim N(0, I)$, can produce extreme values, i.e., $\epsilon$ may reach magnitudes of 6 or -6, because $\epsilon$ is sampled from a standard Gaussian distribution. However, we observe that the token embedding statistics are consistently small across many models (e.g., mean = 0.0179, std = 0.2709). It is challenging for models to adapt to large-value inputs with few fine-tuning epochs. Consequently, we implement noise perturbation on normalized embedding values and subsequently rescale them for model input.
>
>
> ## **9. The possibility of a training-free version of our method**
> Currently, it is challenging to achieve a training-free variant of our method. The difficulty stems from the fact that autoregressive models are not exposed to Gaussian-perturbed tokens during their standard training process. These models are typically optimized for next-token prediction using discrete tokens generated by tokenizer encoder (e.g., VQGAN with CNN encoders), where the inputs are not intentionally corrupted with Gaussian noise throughout pre-training. Consequently, the pre-trained autoregressive models may not encounter noisy token representations, making it difficult to generalize to such inputs without explicit training adaptation.
>
>
> ## **10. About the open-source**
> We will release our code upon paper acceptance.

---

> > ### Comment · Reviewer_ucHV · 2025-08-05
> > **Reply to Authors**
> >
> > Thank you for the detailed response. This answered my questions in detail, and really enhances the work in my opinion.
> >
> > On 6 - if adding the A/B data to the paper or appendix, it would be worth specifying the confidence interval as well, and also looking potentially at per-image-pair agreement across raters. But this already gives a reliable signal that the approach here is if nothing else, quite close in preference and quality to AR, as the preference was not by any means a blowout (and we expect variance between examples just due to the nature of generative models already).
> >
> > On the rate - thank you for the correction, this makes sense!
> >
> > My question on normalization partially stems from related methods in "discrete diffusion" on top of pretrained and from-scratch embeddings (such as SSD-LM or Dirichlet Flow Matching), where even with normalization there can also be reprojection methods, or other things to try and craft the right inputs and outputs. Scaling is also critical and necessary, and the explanation here makes sense, for 8 and 9.

---

> > > ### Author Response · Authors · 2025-08-06
> > > **Reply to Reviewer ucHV**
> > >
> > > Thank you for your valuable feedback. We are glad that the rebuttal has addressed your questions and we appreciate your suggestions. We will include the improvements in our final paper.

---

### Note · Authors · 2025-08-12

We sincerely thank all reviewers and the Area Chair for their invaluable feedback and efforts throughout the review and rebuttal process. Our proposed Speculative Jacobi-Denoising Decoding (SJD2) introduces a new approach to accelerate autoregressive models through parallel token generation, achieving significant speedups while maintaining high-quality generation.

We are encouraged by the positive feedback from reviewers, particularly their recognition of our method’s clarity (Reviewer ucHV), novelty (Reviewer 3CPQ), well-written presentation (Reviewer Q75X) and strong experimental results (Reviewer 3CPQ, Reviewer eMe4). These comments have been instrumental in guiding us to further refine and strengthen our work.


To address the reviewers’ concerns, we made significant efforts in the rebuttal. To expand the evaluation scope (Reviewer Q75X), we conducted additional experiments by applying SJD2 to the advanced Janus-Pro model, comparing SJD2 against other parallel decoding methods and diffusion models, and evaluating Emu3 with SJD2 on the GenEval benchmark. To enhance methodological clarity (Reviewer eMe4), we provided comprehensive explanations of the training and inference procedures, finetuning objective, and timestep injection strategy. We also introduce the prelimiaries for speculative decoding and Jacobi decoding in detail (Reviewer 3CPQ). Additionally, we included an analysis of finetuning details, including wall-clock time and hyperparameters (Reviewer ucHV). Furthermore, we added discussion comparing SJD2 with other methods that integrate autoregressive and diffusion paradigms (Reviewer ucHV, Reviewer eMe4). We also include empirical studies to further investigate the refinement in SJD2 (Reviewer 3CPQ).

We hope these clarifications and additional experiments enhance the paper’s comprehensiveness and accessibility for both reviewers and future readers. We are committed to further refining the manuscript for the camera-ready version.

---

### Decision · Program_Chairs · 2025-09-17

**Decision:**

Accept (poster)

**Comment:**

This paper introduces Speculative Jacobi-Denoising Decoding (SJD2), a novel acceleration framework for autoregressive text-to-image generation. The core scientific claim is that by integrating denoising into Jacobi iterations and introducing a next-clean-token prediction task, SJD2 enables parallel token generation, achieving up to 4x inference speedup while largely preserving visual quality. The paper demonstrates this on multiple recent generative models (Lumina-mGPT, Emu3, Janus-Pro) with both quantitative and qualitative results, including user preference studies. The strengths of the work lie in its clear motivation, methodological novelty, careful experimental validation, and practical acceleration gains. The method is also relatively lightweight, requiring only modest fine-tuning compared to full pretraining.

The weaknesses identified by reviewers include incrementality over prior speculative Jacobi decoding, limited baseline comparisons (initially only against SJD), and questions about generalizability to other AR paradigms (e.g., VAR, MAR). During rebuttal, the authors responded constructively: they added experiments on Janus-Pro, compared against modern speculative/parallel methods (Lantern, ZipAR, Eagle), provided detailed fine-tuning and FLOP/latency analysis, clarified noise segmentation and timestep injection strategies, and conducted user preference A/B testing. The rebuttal substantially mitigated the main criticisms, with reviewers acknowledging improved clarity and breadth of evaluation. Considering the strong technical contribution, convincing empirical validation, and convincing rebuttal, the AC recommends acceptance. The paper makes a solid and timely contribution to the acceleration of autoregressive generative models.